# Breaking Dual Bottlenecks: Evolving Unified Multimodal Models into Self-Adaptive Interleaved Visual Reasoners

**Qingyang Liu** [1 2]  **Bingjie Gao** [1]  **Canmiao Fu** [2]  **Zhipeng Huang** [2]  **Chen Li** [2]  **Feng Wang** [2]  **Shuochen Chang** [1]
**Shaobo Wang** [1]  **Yali Wang** [3]  **Keming Ye** [2]  **Jiangtong Li** [4]  **Li Niu** [1 5]

## Abstract

Recent unified models integrate multimodal understanding and generation within a single framework. However, an "understanding-generation gap" persists, where models can capture user intent but often fail to translate this semantic knowledge into precise pixel-level manipulation. This gap results in two bottlenecks in anything-to-image task (X2I): the **attention entanglement bottleneck**, where blind planning struggles with complex prompts, and the **visual refinement bottleneck**, where unstructured feedback fails to correct imperfections efficiently. In this paper, we propose a novel framework that empowers unified models to autonomously switch between generation strategies based on instruction complexity and model capability. To achieve this, we construct a hierarchical data pipeline that constructs execution paths across three adaptive modes: direct generation for simple cases, self-reflection for quality refinement, and multi-step planning for decomposing complex scenarios. Building on this pipeline, we contribute a high-quality dataset with over 50,000 samples and implement a two-stage training strategy comprising SFT and RL. Specifically, we design step-wise reasoning rewards to ensure logical consistency and intra-group complexity penalty to prevent redundant computational overhead. Extensive experiments demonstrate that our method outperforms existing baselines on X2I, achieving superior generation fidelity among simple-to-complex instructions. The code is released at GitHub.

---

[1]Shanghai Jiao Tong University [2]WeChat Vision, Tencent Inc. [3]Shenzhen Institutes of Advanced Technology, Chinese Academy of Sciences [4]Tongji University [5]miguo.ai. Correspondence to: Jiangtong Li <jiangtongli@tongji.edu.cn>, Li Niu <ustcnewly@sjtu.edu.cn>.

*Proceedings of the $43^{rd}$ International Conference on Machine Learning*, Seoul, South Korea. PMLR 306, 2026. Copyright 2026 by the author(s).

## 1. Introduction

Recent advancements in unified large multimodal models (unified models) (Wu et al., 2025a; Wang et al., 2024b) have integrated multimodal understanding and generation within a single framework. Inspired by the success of Chain-of-Thought (CoT) reasoning, recent studies (Ye et al., 2025d; Han et al., 2026b) have incorporated multimodal CoT reasoning into these unified architectures. This integration enables complex anything-to-image (X2I) tasks (Liu et al., 2025; Niu et al., 2025; Liu et al., 2024; Zhao et al., 2025), requiring models to generate or edit images conditioned on diverse multimodal inputs (*e.g.*, text, layouts, sketches). Although these models demonstrate promising visual reasoning and generative capabilities, a critical disparity remains between their understanding potential and actual generation fidelity (Chen et al., 2025b). Specifically, while current unified models exhibit strong visual comprehension, their ability to execute precise, instruction-following image editing often lags behind. We term this issue the "***understanding-generation gap***", which becomes evident in complex X2I scenarios: the model "***knows***" the user's intent but fails to "***faithfully translate***" this semantic knowledge into accurate pixel-level manipulation.

Driven by this understanding-generation gap, unified models face two primary challenges when executing complex tasks: the attention entanglement bottleneck (Park et al., 2025; Koh et al., 2025) and the visual refinement bottleneck (Wu et al., 2025c). The attention entanglement bottleneck stems from the complexity of user prompts, which act as a barrier to direct, single-pass synthesis and require decomposing instructions into step-by-step edits. Meanwhile, the visual refinement bottleneck arises from unavoidable imperfections in single-step pixel synthesis, creating a need for progressive visual refinement. To mitigate the attention entanglement bottleneck, recent approaches formulate detailed textual plans before execution (Qin et al., 2025; Ye et al., 2025c). However, these methods lack verification mechanisms; the reasoning process remains "blind" to the model's actual generative limits, often yielding unexecutable plans that fail to handle complex, multi-step editing. To handle the visual refinement bottleneck, several meth-

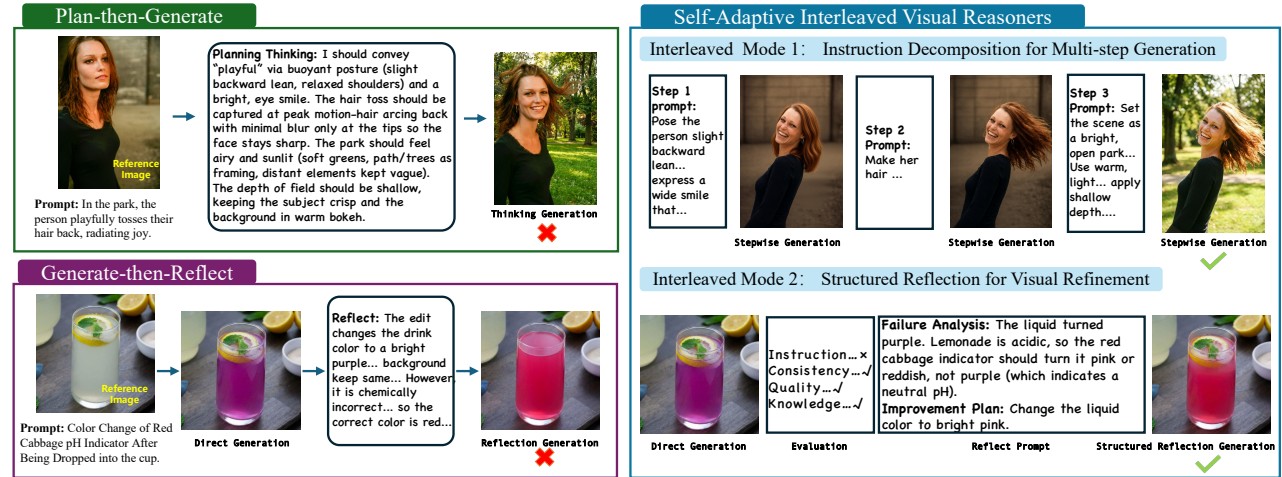

*Figure 1.* Comparison of conventional paradigms and our Self-Adaptive Interleaved Visual Reasoners. **Left:** Existing methods struggle with "blind" planning and unstructured reflection due to understanding-generation gap. **Right:** Our approach adopts adaptive strategies: **Mode 1** decomposes complex prompts via interleaved generation; **Mode 2** employs structured reflection for interleaved visual refinement.

ods employ critiques of initial outputs to guide subsequent refinement (Guo et al., 2025b; Zhuo et al., 2025). These methods typically rely on switching between multiple distinct models, which incurs high computational costs and obstructs automated, progressive self-improvement. Recent efforts attempt to iteratively generate and critique images using a unified model within a feedback loop (Guo et al., 2025a). However, such unstructured critiques mix error analysis and improvement suggestion together, making it difficult to resolve compound errors efficiently. Crucially, no existing work has successfully targeted both bottlenecks simultaneously within a unified framework.

To overcome these two bottlenecks, we propose a Self-adaptive Interleaved Reasoner(InterVR), which equips the unified models with native interleaved generation and dynamic self-evaluation capabilities, allowing it to autonomously switch strategies based on instruction complexity and model capability. Fig. 1 illustrates two interleaved reasoning modes of our reasoner. Specifically, for complex multi-intent instructions, the reasoner utilizes textual planning and interleaved generation to decompose tasks step-by-step. When initial outputs fail to meet quality standards, the reasoner switches to a self-reflection mode for progressive visual refinement. To develop this adaptive capability, we propose a data pipeline driven by a hierarchical escalation mechanism. Given a raw X2I (Qin et al., 2023) input, we first employ the baseline unified models to directly generate the image. We then use an ANALYZER, Qwen3vl-235B (Bai et al., 2025), as an independent critic to evaluate the output's alignment with the user instruction and reference inputs. If the initial generation is unsatisfactory, the ANALYZER diagnoses specific errors in the faulty image and synthesizes a reflection prompt. Guided by this reflection, a GENERATOR, Gemini-3-Pro-Image (Google, 2025),

attempts to rectify the output. If the task remains unresolved after the maximum iteration limit, the ANALYZER identifies reason for such failure. Cases identified as having excessive prompt complexity escalate to the multi-step planning phase, while others (*e.g.*, those requiring specialized knowledge) are filtered out. For multi-step planning phase, the LMM decomposes the complex instruction into sequential sub-tasks, enabling step-by-step generation and intermediate evaluation until the objective is met. Following strict human verification, we obtain a high-fidelity interleaved dataset with over 50,000 samples that integrates generation, reflection, planning, and evaluation across three adaptive modes.

Leveraging this constructed dataset, we implement a two-stage training paradigm. First, we apply Supervised Fine-Tuning (SFT) to adapt the model to the interleaved reasoning syntax and basic instruction following. Subsequently, we employ Reinforcement Learning (RL) with Group Relative Policy Optimization algorithm (GRPO) to optimize the model's strategic planning, ensuring it selects the most effective generation mode for varying instruction complexities. Beyond standard format and outcome rewards assessed by an LMM, we introduce a step-wise reasoning reward and an intra-group complexity penalty. The step-wise reasoning reward verifies the logical consistency of intermediate steps, utilizing the LMM to validate each textual output. To prevent inefficient over-reasoning, the intra-group complexity penalty favors simpler generation modes when they yield results equivalent to complex ones, thereby balancing generation quality with computational efficiency. Extensive experiments and visualization results confirm that our method outperforms existing baselines (EUM3.5 (Cui et al., 2025), *etc.*) among text-to-image, image editing, and X2I tasks. Our contributions can be summarized as:

- We propose a unified framework with interleaved plan or reflect, resolving the understanding-generation gap of unified models by addressing attention entanglement and visual refinement bottlenecks.

- We design a hierarchical data synthesis pipeline that constructs execution paths across direct, reflective, and multi-step planning modes. Based on this pipeline, we contribute a large-scale high-fidelity dataset to enable adaptive X2I generation.

- We implement a two-stage training strategy with SFT and RL. With our proposed step-wise reasoning reward and intra-group complexity penalty, we ensure logical correctness while minimizing computational costs.

## 2. Related Work

### 2.1. Unified Large Multimodal Models

The field of unified multimodal understanding and generation (Pan et al., 2025; Shi et al., 2024; Sun et al., 2023; Ge et al., 2024; Liao et al., 2025b; Chang et al., 2026) has advanced rapidly in recent years. Current unified models generally fall into three categories: diffusion-based (Swerdlow et al., 2025; Li et al., 2025c; Yang et al., 2025), autoregressive (Wu et al., 2025e; Li et al., 2025a), and hybrid autoregressive–diffusion architectures (Xie et al., 2024; Zhou et al., 2024; Ma et al., 2025c). Diffusion-based models generate images by iterative denoising, while autoregressive models predict visual and textual tokens sequentially. Hybrid architectures combine autoregressive reasoning with diffusion decoding for structured planning and high-quality synthesis. However, unified models still suffer from an understanding–generation gap, where semantic understanding does not reliably translate into pixel-level outputs. This gap creates two bottlenecks in X2I tasks: the **attention entanglement bottleneck**, where blind planning fails on complex prompts, and the **visual refinement bottleneck**, where unstructured feedback cannot efficiently correct errors. Thus, a unified solution is needed to tightly couple reasoning and generation for adaptive instruction decomposition and output refinement.

### 2.2. Reasoning in Generation

Recent researches increasingly focus on the reasoning and reflection capabilities of LMMs (Wang et al., 2024a; Hurst et al., 2024; Liu et al., 2026b; Peng et al., 2026; Ma et al., 2025b; Han et al., 2026a). Beyond their success in multimodal comprehension, studies now apply these capabilities to enhance image/video generation and editing (Han et al., 2025; Fang et al., 2025; Huang et al., 2025; Jiang et al., 2025b; Liu et al., 2026a; Chen et al., 2025a; Ye et al., 2025b; Wu et al., 2025f; Han et al., 2024). LMMs are employed to analyze intermediate results, identifying inconsistencies or instruction violations to guide subsequent generation steps.

These reasoning-driven pipelines allow models to refine outputs through self-evaluation, improving instruction adherence and visual fidelity in complex scenarios. Recently, this strategy has been integrated into unified models (Yin et al., 2025), which jointly support understanding and generation within a single framework. By aligning semantic reasoning with generation in a unified framework, unified models incorporate planning and corrective feedback directly into the generation process. Many reasoning-in-generation methods use RL (Shao et al., 2024; Rafailov et al., 2023) with self-evaluation or external feedback as rewards, making reasoning trainable for better instruction following and iterative correction. However, existing approaches still fail to close the understanding–generation gap, as they mostly follow rigid "Plan-then-Generate" (Jiang et al., 2025a; Liao et al., 2025a; Gao et al., 2025a) or "Generate-then-Reflect" (Li et al., 2025b; Gao et al., 2025b) pipelines. Lacking adaptive control over when to plan or reflect, they often incur blind planning or inefficient feedback, and thus cannot simultaneously resolve attention entanglement and visual refinement bottlenecks in complex X2I tasks.

## 3. Data Construction

To equip the unified model with broad generalization capabilities, we construct a dataset comprising diverse source modalities paired with user instructions of varying complexity. To enable the unified model to switch between generation, reflection, and planning, we develop an automated, hierarchical data construction pipeline. This pipeline employs the ANALYZER (Qwen-235B) as an evaluator, failure diagnostician, and task planner, alongside the GENERATOR (Gemini-3-Pro-Image), which synthesizes images based on reflection prompts or step-wise instructions. The construction process automatically categorizes data into three operational modes according to instruction complexity. Fig. 2 illustrates representative cases from each mode, demonstrating the effectiveness of our data construction pipeline.

### 3.1. Direct Generation

We initiate the process by prompting the baseline unified model for direct generation, after which the ANALYZER evaluates the output across four criteria: **1. Instruction:** Verifies whether the user instructions are accurately executed. **2. Consistency:** Checks if the identity and attributes of unedited regions in the reference image remain preserved. **3. Quality:** Assesses overall visual fidelity, identifying potential artifacts or degradation. **4. Knowledge:** Confirms that the generated content aligns with physical laws and commonsense. If the output meets all criteria, we retain the trajectory as a direct generation instance, representing scenarios where single-step inference suffices. For the direct generation mode, each data sample consists of the generated image $G_1$ and its corresponding evaluation $E_1$.

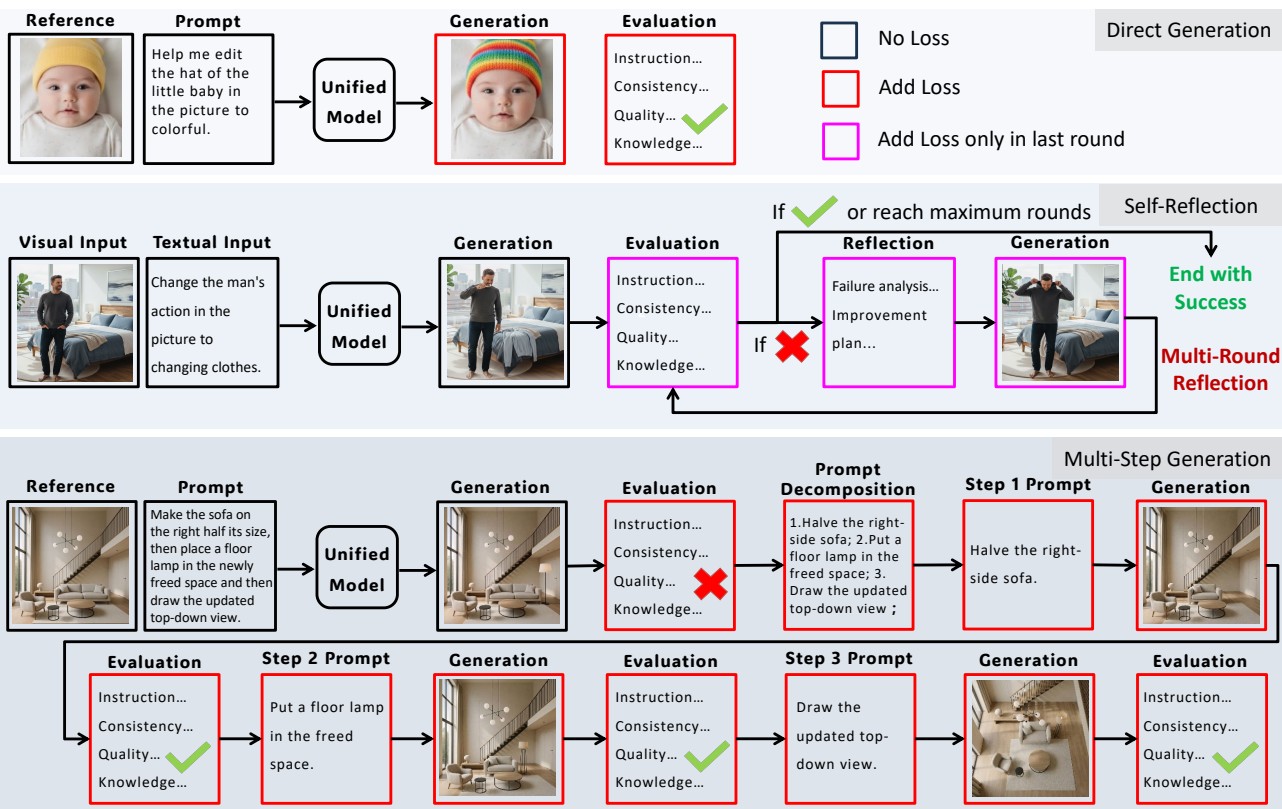

*Figure 2.* Illustration of the three kinds of training data in our dataset and the selective loss masking strategy for in the SFT stage.

## 3.2. Self-Reflection

If the initial generation fails validation, the pipeline initiates a self-correction loop, limited to a maximum of three iterations. The ANALYZER processes the original instruction, the reference image, the evaluation text and the rejected output to formulate a targeted "reflection prompt." This prompt consists of two key components: a failure analysis that identifies specific errors, and an improvement plan that guides the subsequent generation. Guided by this reflection, the GENERATOR attempts to correct the image errors. Once a revised output meets the ANALYZER's evaluation standards, the iterative loop terminates. We then record the successful trajectory as a self-reflection data instance, indicating that the self-correction is sufficient to resolve the failure. Each data sample for self-reflection is defined as $\bigcup_{i=1}^{K-1}\{G_i, E_i, R_i\} \cup \{G_K, E_K\}$, where $G_i$, $E_i$, and $R_i$ denote the generated image, evaluation, and reflection at the $i$-th iteration, respectively, and $K$ represents the total number of iterations.

## 3.3. Multi-step Generation

If the task remains unsolved after reaching the maximum three-iteration limit, the pipeline avoids automatic escalation. Instead, the ANALYZER performs a comprehensive diagnosis to identify the root cause of the failure. Specifi-

cally, if the failure stems from excessive prompt complexity, the pipeline escalates to explicit task decomposition; otherwise, cases attributed to other factors (*e.g.*, lack of domain knowledge) are filtered out. In this phase, the ANALYZER acts as a planner to break down the original instruction into sequential sub-tasks, which the GENERATOR executes one by one while the ANALYZER evaluates each intermediate step result. Upon successful completion of the sequence, we perform trajectory pruning, filtering out the previously failed reflection attempts. This process yields a clean sequential reasoning path, constituting the Multi-step Generation data. We archive this pruned trajectory as a multi-step instance, representing the explicit sequential task decomposition are necessary to resolve complex instructions. Each data sample for multi-step generation is defined as $\{G_1, E_1\} \cup \bigcup_{i=2}^{N+1}\{S_i, G_i, E_i\}$, where, $G_1$ is the direct generation and $E_1$ is its failure analysis; $S_i$, $G_i$, and $E_i$ represent the sub-instruction, generated image, and evaluation at sub-step $i-1$, with $N$ denoting the number of planning steps for the multi-step generation.

## 3.4. Human Verification

To guarantee the high quality of our hybrid dataset, every synthesized instance undergoes a strict human verification process. For each data sample, we employ two human annotators to review both the final generation outcomes

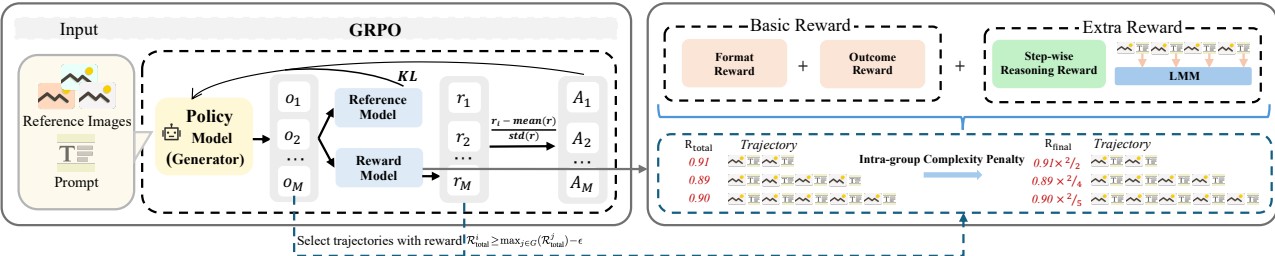

*Figure 3.* Overview of the GRPO-based RL stage. **Left:** The Policy Model generates a group of candidate trajectories ($o_1 \ldots o_M$) to compute group-relative advantages ($A_i$). **Right:** The composite reward function aggregates **Basic Rewards** (Format, Outcome) and an **Extra Reward** (Step-wise Reasoning) validated by an LMM. An **Intra-group Complexity Penalty** modulates the Total Reward ($R_{\text{total}}$): among successful trajectories, those with redundant steps are scaled down to derive the Final Reward ($R_{\text{final}}$), favoring effective solution.

and the logical coherence of all intermediate reasoning and planning steps. Only instances accepted by all annotators are retained as valid cases. Through this combination of automated construction and manual quality assurance, the resulting dataset enables the model to perform iterative self-reflection and multi-step decomposition, grounding high-fidelity generation in text-based reasoning signals.

# 4. Training

We employ a two-stage training strategy to equip the unified model with adaptive reasoning skills and computational efficiency. In the first stage, supervised fine-tuning adapts the model to the interleaved format, enabling it to execute textual planning and visual reflection during generation. In the second stage, we explore reinforcement learning with GRPO to optimize the unified model, training the model to select the most efficient execution path for a given task.

## 4.1. Supervised Fine-Tuning

In the SFT stage, we train the unified model using our curated dataset, which covers all three operational modes. Standard training on this dataset would require the model to predict suboptimal intermediate images, potentially degrading its generative fidelity. To prevent this, we employ a selective loss masking strategy. As shown in Fig. 2, given an instruction-reference pair $c$, the model minimizes the standard auto-regressive negative log-likelihood loss, computed on a selected subset of the output sequence $\mathcal{O}$:

$$\mathcal{L}_{\text{SFT}} = -\sum_{t \in \mathcal{O}} \log P(x_t | x_{<t}, c), \qquad (1)$$

where $x_t$ represents the $t$-th token (text or image patch). The target subset $\mathcal{O}$ is defined according to the data mode.
**Direct Generation:** For direct execution, the loss is applied to both the generated image $G_1$ and its evaluation $E_1$, such that $\mathcal{O}_{\text{direct}} = \{G_1, E_1\}$.
**Self-Reflection:** Given a reflection trajectory with $k$ attempts (where only the $K$-th succeeds), we ensure the model learns correction logic without memorizing the artifacts of

failed generations. Therefore, we mask the loss for all prior failed generations $\{G_1, ..., G_{K-1}\}$ and their evaluations up to step $K-2$. The loss is applied only to the final correction phase: the diagnosis of the last failure $E_{K-1}$, the reflection prompt $R_{K-1}$ (containing analysis and planning), and the final successful generation $G_K$ with its evaluation $E_K$, defined as $\mathcal{O}_{\text{reflect}} = \{E_{K-1}, R_{K-1}, G_K, E_K\}$.
**Multi-step Generation:** To learn explicit task decomposition, we compute loss on the evaluation of the initial direct attempt $E_1$, followed by the complete multi-step planning sequence. For each sub-step $i - 1$, the loss includes the sub-instruction $S_i$, the intermediate generation $G_i$, and its validation $E_i$, given by $\mathcal{O}_{\text{multi}} = \{E_1\} \cup \bigcup_{i=2}^{N+1} \{S_i, G_i, E_i\}$. Through this masking strategy, the unified model learns self-reflection and multi-step decomposition while avoiding training on low-quality visual intermediates.

## 4.2. Reinforcement Learning Stage

While SFT establishes the foundational skills for interleaved planning and reflection, we aim to enhance this framework, enabling the model to dynamically select the most efficient solution. To this end, we introduce a second training stage based on GRPO. We curate a specialized RL dataset containing 50,000 high-quality samples selected from UnicEdit-10M (Ye et al., 2025b), X2Edit (Ma et al., 2025a), AnyEdit (Yu et al., 2025), Pick-a-Pic (Kirstain et al., 2023), and UltraEdit (Zhao et al., 2024), covering a wide range of task complexities. To guide policy optimization, as shown in Fig. 3, we define a composite reward function assessing three key aspects: outcome rewards for generation fidelity, format rewards for structural validity, and step-wise reasoning rewards for logical consistency. Rather than treating efficiency as an independent reward, we introduce an Intra-group Complexity Penalty to regulate the generation process, which dynamically adjusts the total reward, ensuring that high-fidelity interleaved reasoning does not lead to inefficient "over-reasoning."

**Outcome Reward:** Aligning with the assessment criteria in Section 3, the outcome reward evaluates the final generated image. It is calculated as a normalized weighted sum of

*Table 1.* Quantitative evaluation results on the KRIS-Bench benchmark. Best performance are highlighted **bolded**.

| | Models | Factual | Conceptual | Procedural | Overall |
|---|---|---|---|---|---|
| **Close-source Model** | Gemini 2.5 flash | 77.03 | 78.29 | 75.93 | 77.29 |
| | Doubao | 78.10 | 76.86 | 76.93 | 77.31 |
| | GPT4o | 79.80 | 81.37 | 78.32 | 80.09 |
| **Open-Source Model** | OmniGen2 (Wu et al., 2025c) | 57.36 | 44.20 | 47.79 | 49.71 |
| | BAGEL-thinking (Deng et al., 2025a) | 66.18 | 61.92 | 49.02 | 60.18 |
| | BAGEL (Deng et al., 2025b) | 60.26 | 55.86 | 51.69 | 56.21 |
| | Uniworld-V1 (Lin et al., 2025) | 47.71 | 44.80 | 47.92 | 50.27 |
| | Flux-Kontext-dev (Labs et al., 2025) | 53.28 | 50.36 | 42.53 | 49.54 |
| | UniWorld-FLUX.1-Kontext-Dev (Li et al., 2025d) | 55.50 | 51.39 | 43.76 | 51.04 |
| | UniWorld-Qwen-Image-Edit (Li et al., 2025d) | 61.72 | 56.38 | 46.69 | 55.98 |
| | Step1X-Edit v1.1 (Liu et al., 2025) | 53.05 | 54.34 | 44.66 | 51.59 |
| | Qwen-Image-Edit-2509 (Wu et al., 2025b) | 61.47 | 56.79 | 47.07 | 56.15 |
| | ReasonEdit-Q (thinking+reflection) (Yin et al., 2025) | 63.92 | 64.85 | 52.41 | 61.57 |
| | Emu3.5 (Cui et al., 2025) | 78.59 | 71.92 | 71.14 | 73.75 |
| | **Ours** | **84.24** | **74.83** | **85.53** | **80.18** |

four metrics: instruction following $S_{\text{instr}}$, consistency $S_{\text{cons}}$, quality $S_{\text{qual}}$, and knowledge $S_{\text{know}}$:

$$\mathcal{R}_{\text{o}} = w_1 S_{\text{instr}} + w_2 S_{\text{cons}} + w_3 S_{\text{qual}} + w_4 S_{\text{know}} \quad (2)$$

where $w_1, w_2, w_3, w_4$ are fixed normalization factors rather than tunable hyperparameters. Each sub-score is measured on a 1–5 scale and rescaled by multiplying by $0.2$ to map it to the range $[0, 1]$. These normalized scores are then averaged across the four dimensions, resulting in $w_1 = w_2 = w_3 = w_4 = 0.05$. Thus, Eq. (2) serves only to define the normalized outcome reward without introducing additional variables for reward tuning.

**Format Reward:** To ensure the model adheres to the defined reasoning structure during RL exploration, we implement a binary validity check:

$$\mathcal{R}_{\text{f}} = \begin{cases} 1, & \text{if trajectory structure is valid,} \\ 0, & \text{otherwise.} \end{cases} \quad (3)$$

**Step-wise Reasoning Reward:** To mitigate the credit assignment bottleneck in long-horizon generation, we introduce a dense reasoning reward. The ANALYZER evaluates the logical validity of intermediate textual outputs, including failure analysis, reflection prompts, and step decomposition. For a trajectory with $T$ intermediate reasoning steps:

$$\mathcal{R}_{\text{s}} = \frac{1}{T} \sum_{t=1}^{T} \text{ANALYZER}(\text{text}_t), \quad (4)$$

where $\text{ANALYZER}(\text{text}_t) \in [0, 1]$ represents the validity score of the textual output at step $t$.

**Total Reward:** We aggregate these components into a weighted total reward: $\mathcal{R}_{\text{total}} = \alpha_1 \mathcal{R}_{\text{o}} + \alpha_2 \mathcal{R}_{\text{f}} + \alpha_3 \mathcal{R}_{\text{s}}$, where $\alpha_1, \alpha_2$, and $\alpha_3$ balance the final generation fidelity, structural correctness, and intermediate reasoning logic.

**Intra-group Complexity Penalty:** To prevent inefficient "over-reasoning," we introduce a complexity penalty that favors the simplest effective path. Within a sampled group $G$, we identify a set of competitive trajectories that achieve rewards within a margin $\epsilon$ of the maximum performance. Let $N_{\text{img}}^*$ denote the minimum number of generated images among competitive trajectories. For any trajectory $i \in G$, we adjust its total reward based on its image count $N_{\text{img}}^i$:

$$\mathcal{R}_{\text{final}}^i = \begin{cases} \mathcal{R}_{\text{total}}^i + \frac{N_{\text{img}}^*}{N_{\text{img}}^i}, & \text{if } \mathcal{R}_{\text{total}}^i \geq \max_{j \in G}\left(\mathcal{R}_{\text{total}}^j\right) - \epsilon, \\ \mathcal{R}_{\text{total}}^i, & \text{otherwise.} \end{cases}$$

$$(5)$$

This mechanism ensures that among high-performing solutions, the model favors those with lower computational cost.

## 5. Experiments

### 5.1. Experiment Setup

**Comparison Baselines.** We adopt Emu3.5 (Cui et al., 2025) as our backbone unified model. To validate the effectiveness of our adaptive framework, we compare it against the vanilla Emu3.5, which serves as the direct generation baseline. Furthermore, we benchmark against two fixed reasoning strategies of unified model: (1) Plan-then-Generate (Jiang et al., 2025a; Liao et al., 2025a), employing a static textual planning stage before execution; and (2) Generate-then-Reflect (Li et al., 2025b; Qin et al., 2025), relying on iterative visual critique for refinement.

**Evaluation Dataset.** We assess performance across diverse generation and editing scenarios using three standard benchmarks. For standard Text-to-Image (T2I) generation, we use GenEval (Ghosh et al., 2023) to measure general generation quality and semantic alignment. For Image Editing, we employ KRIS-Bench (Wu et al., 2025g) to evaluate advanced

*Table 2.* Ablation study on reasoning modes and training stages. **Top:** Impact of distinct data modes evaluated on a 30k subset. **Bottom:** Effectiveness of RL components on the full 50k dataset, reporting generation efficiency (Avg. Imgs).

| Setting | GenEval | KRIS | Omni | Avg. Imgs |
|---|---|---|---|---|
| *Impact of Reasoning Modes (30k subset)* | | | | |
| Direct Only | 0.86 | 75.16 | 8.89 | - |
| w/o Multi-step | 0.87 | 77.24 | 8.95 | - |
| w/o Reflection | 0.86 | 75.21 | 9.03 | - |
| Full Mix (Balanced) | **0.88** | **78.24** | **9.15** | - |
| *Impact of RL Training (50k full set)* | | | | |
| SFT Only | 0.86 | 79.16 | 9.12 | 2.45 |
| w/o Step-wise Reward | 0.88 | 79.65 | 9.25 | 1.62 |
| w/o Complexity Penalty | 0.89 | 80.25 | 9.38 | 2.73 |
| SFT + RL (**Ours**) | **0.89** | **80.18** | **9.35** | **1.56** |

reasoning and the interpretation of abstract instructions. Finally, for complex Anything-to-Image (X2I) tasks, we select OmniContext (Wu et al., 2025c) to assess subject-driven and multi-reference generation across varying granularities. Further details regarding training and evaluation settings are provided in Appendix A.

### 5.2. Experimental Results

**GenEval:** As shown in Table 5, our method achieves a leading score of 0.89, outperforming Emu3.5 (0.86), VA-CoT (0.84), and FLUX.1-dev (0.82). The results show that our self-adaptive interleaved reasoning effectively improves spatial and compositional understanding, especially in reasoning-intensive categories such as Counting, Position, and Color Attribution.

**KRIS-Bench:** As presented in Table 1, our method attains the best overall score of 80.18, surpassing all open-source baselines and even proprietary models. In particular, it achieves a large gain in Procedural Knowledge (85.53 vs. 71.14 for Emu3.5) and strong performance in Factual Knowledge (84.24), demonstrating that interleaved planning and self-reflection effectively enhance multi-step instruction following and semantic correction.

**OmniContext:** As detailed in Table 3, our method sets a new state of the art with an average score of 9.35, outperforming Emu3.5 (8.82), VACoT (8.26), GPT-4o (8.80), and Gemini-2.5-Flash (7.84). Notably, it performs best in the "Multiple" and "Scene" categories which involve complex multi-subject interactions, indicating that multi-step planning can effectively disentangle multi-subject interactions and reduce identity mixing and attribute leakage.

### 5.3. Ablation Studies

Table 2 presents an ablation analyzing the impact of reasoning modes and the contributions of RL components.
**Impact of reasoning modes.** We evaluate the contribution

of distinct reasoning strategies using a controlled 30k data subset. We compare the "Full Mix" strategy against two ablated settings: "w/o Multi-step" and "w/o Reflection" . The "Direct Only" baseline yields the poorest performance across all metrics (e.g., 75.16 on KRIS-Bench), demonstrating that data scale alone is insufficient without structured reasoning. Removing reflection data ("w/o Reflection") leads to a significant drop in KRIS-Bench (75.21 vs. 78.24), highlighting the necessity of self-correction for visual refinement. Conversely, excluding multi-step planning ("w/o Multi-step") impairs performance on OmniContext (8.95 vs. 9.15), confirming that explicit task decomposition is required for complex, multi-subject scenarios. "Full Mix" setting achieves the highest scores across all benchmarks, verifying the complementary of planning and reflection modes.
**Effectiveness of RL training.** Comparing the "SFT Only" baseline with our final "SFT + RL" model, we observe consistent gains (*e.g.*, OmniContext 9.12 → 9.35) alongside a sharp reduction in average generated images (2.45 → 1.56). Removing the step-wise reasoning reward results in performance declines across all benchmarks (*e.g.*, KRIS-Bench falls to 79.65), indicating that dense LMM supervision is vital for ensuring logical consistency. Finally, eliminating the intra-group complexity penalty causes the average image count to surge to 2.73; while this yields marginally higher scores through excessive trial-and-error, it introduces severe computational redundancy, whereas our full model balances high fidelity with optimal efficiency.

## 6. Case Study

Figure 4 illustrates how our adaptive reasoner resolves complex failures by actively leveraging multimodal understanding during generation. In the top logic-driven case, the base model fails to recognize the prime number sequence (5, 7, 11, 13, 17, ?) and instead produces a visually plausible but logically incorrect "1?". The **Reflection Mode** bridges this understanding–generation gap by explicitly diagnosing the failure as a "reasoning error" rather than a visual artifact, thereby recovering the missing semantic constraint. Leveraging this insight, the analyzer then guides the generator to render the mathematically correct "19". In the bottom compositional case, the **Multi-Step Mode** handles a dense prompt with interacting elements such as "keys" and "broken blinds". While standard generation may introduce consistency errors, such as hallucinating a second clock or morphing keys into coins, our step-wise evaluator serves as a semantic guardrail against such failures. By flagging these artifacts (marked with ×) against the global context, the model prevents error propagation and maintains object permanence throughout the long-horizon generation process. These examples highlight that many X2I failures arise from a lack of explicit reasoning over compositional constraints. **More visualizations are provided in the Appendix.**

*Table 3.* Quantitative comparison results on OmniContext benchmark. 'Char.' and 'Obj.' denote Character and Object, respectively.

| Model | SINGLE | | MULTIPLE | | | SCENE | | | Average↑ |
|---|---|---|---|---|---|---|---|---|---|
| | Char. | Obj. | Char. | Obj. | C. + O. | Char. | Obj. | C. + O. | |
| OmniGen (Xiao et al., 2025) | 7.21 | 5.71 | 5.65 | 5.44 | 4.68 | 3.59 | 4.32 | 5.12 | 4.34 |
| UNO (Wu et al., 2025d) | 6.60 | 6.83 | 2.54 | 6.51 | 4.39 | 2.06 | 4.33 | 4.37 | 4.71 |
| BAGEL (Deng et al., 2025b) | 5.48 | 7.03 | 5.17 | 6.64 | 6.24 | 4.07 | 5.71 | 5.47 | 5.73 |
| OmniGen2 (Wu et al., 2025c) | 8.05 | 7.58 | 7.11 | 7.13 | 7.45 | 6.38 | 6.71 | 7.04 | 7.18 |
| Qwen-Image-Edit (Wu et al., 2025b) | 8.35 | 9.13 | 7.65 | 8.85 | 7.90 | 5.16 | 7.75 | 6.73 | 7.69 |
| Gemini 2.5 Flash (Sep. 2025) | 8.62 | 8.91 | 7.88 | 8.92 | 7.39 | 7.29 | 7.05 | 6.68 | 7.84 |
| Uni-CoT (Qin et al., 2025) | - | - | 7.12 | 8.84 | 7.97 | 7.07 | 8.16 | 8.20 | 7.89 |
| Echo-4o (Ye et al., 2025a) | - | - | 8.07 | 7.50 | 8.29 | 8.62 | 8.00 | 8.08 | 8.09 |
| VACoT (Ye et al., 2025d) | - | - | 7.82 | 9.21 | 8.30 | 7.55 | 8.67 | 7.99 | 8.26 |
| GPT4o (Sep. 2025) | 8.90 | 9.01 | 9.07 | 8.95 | 8.54 | 8.90 | 8.44 | 8.60 | 8.80 |
| Emu3.5 (Cui et al., 2025) | 8.72 | 9.46 | 8.65 | 9.09 | 8.78 | 8.78 | 8.89 | 8.15 | 8.82 |
| **Ours** | **9.40** | **9.50** | **9.56** | **9.22** | **9.44** | **9.56** | **9.22** | **8.86** | **9.35** |

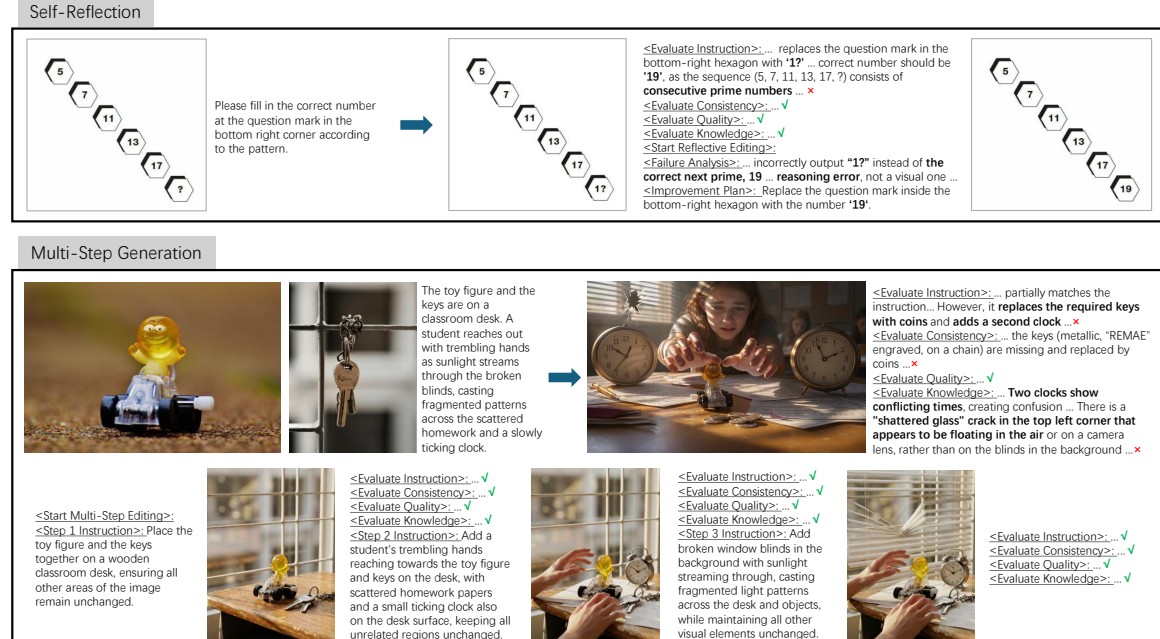

*Figure 4.* Qualitative examples of our adaptive reasoning. Top: Self-Reflection corrects logical errors in a visual puzzle. Bottom: Multi-Step Planning detects and prevents object inconsistency and implausibility during complex scene synthesis.

## 7. Conclusion

In this work, we address the understanding-generation gap in unified models by proposing an adaptive framework that dynamically switches between direct generation, interleaved planning, and self-reflection. To support this, we construct a hierarchical reasoning dataset and implement a two-stage training strategy combining SFT with GRPO, utilizing step-wise reasoning rewards and complexity penalties to optimize decision-making. Extensive experiments demonstrate that our method achieves state-of-the-art performance on benchmarks like GenEval and KRIS-Bench, resolving the dual bottlenecks of blind planning and unstructured feedback while maintaining high inference efficiency. Our findings suggest that empowering unified models with autonomous strategic selection offers a scalable path toward more reliable and interpretable multimodal generation.

## Impact Statement

This paper presents work whose goal is to advance the field of machine learning. There are many potential societal consequences of our work, none of which we feel must be specifically highlighted here.

## Acknowledgement

The work was supported by the National Natural Science Foundation of China (Grant No. 62402341, Grant No. 62471287).

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

This appendix provides supplementary details, analyses, and experiments to support our main findings. The contents are organized as follows:

- Appendix A details the implementation specifics for the two-stage fine-tuning and outlines our experimental settings.
- Appendix B provides comprehensive statistics regarding the constructed dataset.
- Appendix C includes supplementary evaluations on the GenEval benchmark.
- Appendix D showcases additional qualitative examples demonstrating the effectiveness of our method.
- Appendix E lists the full set of prompts employed during data annotation and the RL stage.

## A. Implementation Details

We use a internal proprietary distributed infrastructure for the two-stage training and inference. All models in the main submission are initialized from **EUM 3.5**. The hyperparameter settings for each stage are listed in Table 4.

*Table 4.* Main hyperparameter settings for SFT and RL training stages.

| Parameter | SFT | RL |
|---|---|---|
| resolution | 512×512 | 512×512 |
| train_batch_size | 128 | 128 |
| rollout_size | - | 8 |
| num_epochs | 1 | 1 |
| optimizer | adamw_bf16 | adamw_bf16 |
| temperature | - | 1 |
| kl_coef | - | 1e-2 |
| learning_rate | 1e-5 $\rightarrow$ 1e-6 | 1e-6 |
| scheduler | Cosine | Cosine |
| warm-up ratio | 0.1 | 0.1 |
| Outcome Reward Weight ($\alpha_1$) | - | 0.7 |
| Format Reward Weight ($\alpha_2$) | - | 0.1 |
| Step-wise Reasoning Reward Weight ($\alpha_3$) | - | 0.2 |
| Intra-group Complexity Penalty ($\epsilon$) | - | 0.05 |

## B. Dataset Statistics

Our dataset comprises 50,000 samples distributed across three operational modes: **Direct Mode** (10,000 samples), **Reflection Mode** (20,000 samples), and **Multi-step Mode** (20,000 samples). These modes address varying levels of reasoning complexity and execution depth. The data is organized into six primary dimensions, spanning 21 sub-categories:

- **Object Manipulation**: Basic modifications including *Subject Addition*, *Removal*, *Replacement*, and *Part Completion*.
- **Attribute Modification**: Granular adjustments such as *Color*, *Material*, *Size*, *Count*, and *Anomaly Correction*.
- **Spatial & Viewpoint**: Geometric reasoning including *Viewpoint Change*, *Pose Alteration*, and *Spatial Arrangement*.
- **Global & Style**: Holistic alterations such as *Background Change*, *Style Transfer*, and *Tone/Lighting Adjustment*.
- **Dynamics & Logic**: High-level semantic tasks involving *Motion Change*, *Temporal Evolution*, and *Text Modification*.
- **Multi-Image Operations**: Inter-image tasks such as *Composition*, *Object Replacement*, and *Reference Transfer*.

## C. Extra Experiments

As detailed in Table 5, our method secures the leading position among unified models with an overall score of 0.89, outperforming both the strong baseline Emu3.5 (0.86) and recent CoT-based approaches such as VACoT (0.84). These results underscore the effectiveness of our framework in closing the understanding-generation gap. While standard unified models often struggle to translate complex semantic relationships into precise pixel arrangements, our approach exhibits marked improvements in spatial and compositional tasks. Specifically, in reasoning-intensive categories, we outperform VACoT by notable margins: Counting (0.90 *v.s.* 0.80), Position (0.79 *v.s.* 0.66), and Color Attribution (0.81 *v.s.* 0.71).

*Table 5.* Evaluation of text-to-image generation ability on GenEval benchmark. Best performance are highlighted **bolded**.

| | Model | Single Obj. | Two Obj. | Counting | Colors | Position | Color Attri. | Overall |
|---|---|---|---|---|---|---|---|---|
| **Gen-only** | Emu3-Gen (Wang et al., 2024b) | 0.98 | 0.71 | 0.34 | 0.81 | 0.17 | 0.21 | 0.54 |
| | SDXL (Podell et al., 2023) | 0.98 | 0.74 | 0.39 | 0.85 | 0.15 | 0.23 | 0.55 |
| | DALL-E 3 (Betker et al., 2023) | 0.96 | 0.87 | 0.47 | 0.83 | 0.43 | 0.45 | 0.67 |
| | SD3-Medium (Esser et al., 2024) | 0.99 | 0.94 | 0.72 | 0.89 | 0.33 | 0.60 | 0.74 |
| | FLUX.1-dev (Labs, 2024) | 0.98 | 0.93 | 0.75 | **0.93** | 0.68 | 0.65 | 0.82 |
| **Unified Model** | TokenFlow-XL (Qu et al., 2025) | 0.95 | 0.60 | 0.41 | 0.81 | 0.16 | 0.24 | 0.55 |
| | Show-o (Xie et al., 2024) | 0.98 | 0.80 | 0.66 | 0.84 | 0.31 | 0.50 | 0.68 |
| | Janus-Pro-7B (Chen et al., 2025c) | 0.99 | 0.89 | 0.59 | 0.90 | **0.79** | 0.66 | 0.80 |
| | MetaQuery-XL (Pan et al., 2025) | - | - | - | - | - | - | 0.80 |
| | BAGEL (Deng et al., 2025b) | 0.99 | 0.92 | 0.78 | 0.87 | 0.53 | 0.64 | 0.79 |
| | UiG (Lyu et al., 2025) | 0.99 | 0.92 | 0.81 | 0.89 | 0.61 | 0.69 | 0.82 |
| | Uni-CoT (Qin et al., 2025) | 0.99 | **0.95** | 0.82 | 0.89 | 0.60 | 0.72 | 0.83 |
| | VACoT (Ye et al., 2025d) | 0.99 | **0.95** | 0.80 | 0.90 | 0.66 | 0.71 | 0.84 |
| | Emu3.5 (Cui et al., 2025) | - | - | - | - | - | - | 0.86 |
| | **Ours** | 0.98 | 0.94 | **0.90** | **0.93** | **0.79** | **0.81** | **0.89** |

Notably, our unified model even surpasses specialized text-to-image generators like FLUX.1-dev (0.82) and SD3-Medium (0.74), suggesting that interleaved reasoning is essential for high-fidelity semantic alignment. This indicates that the self-adaptive mechanism, dynamically decomposing prompts or refining layouts, successfully handles compositional constraints that limit static single-pass or standard CoT baselines.

## D. Case Study

Figure 5 illustrates a counting-based editing task: "Remove two computers" from a five-monitor setup. The baseline Emu3.5 fails to initiate the edit, leaving the scene unchanged due to a failure in instruction grounding. In contrast, our model initially over-executes, removing all five monitors. The self-reflection mode detects this quantity mismatch, diagnosing a misinterpretation of the instruction's scope. By formulating a specific spatial plan, "Remove only the leftmost and right-most monitors", the system guides the generator to retain the three central screens, achieving the precise quantitative edit.

Figure 6 illustrates a case requiring domain-specific physical understanding: depicting oxygen release during algae photosynthesis. The base model defaults to superficial visual metaphors, rendering glowing blue sparkles and textual labels. This represents a misrepresentation of scientific concepts, where the model prioritizes abstract semantic associations over physical reality. The reflection module identifies this conceptual error, critiquing the lack of buoyancy, refraction, and biological accuracy. By enforcing the correct physical properties, the system guides the generator to replace artificial effects with transparent, ascending bubbles, ensuring the output aligns with scientific principles rather than abstract visual tropes.

Figures 7 and 8 illustrate a complex multi-reference task: integrating three specific identities into a unified wedding scene. Direct Generation and baseline Emu3.5 suffer from severe attention entanglement under these multi-subject constraints. Specifically, direct generation fails to isolate identity inputs from the scene context, leading to counting errors (*e.g.*, hallucinating a fourth individual). In contrast, our **Multi-Step Mode** autonomously decomposes the task into sequential sub-goals: establishing spatial layout, refining clothing attributes, and synthesizing the background. This hierarchical execution ensures strict identity preservation and correct spatial arrangement, effectively resolving the counting and consistency issues that limit baseline models.

Figures 9 and 10 illustrate a spatial reasoning task requiring a brown bear to be positioned "several feet away" from a dining table. Direct generation methods (Base, Emu3.5) fail to adhere to this constraint due to attention entanglement, tending to cluster objects in the foreground and placing the bear immediately adjacent to the table. In contrast, our **Multi-Step Mode** resolves this by decomposing scene construction: it first generates the table to establish the spatial ground plane, then conditions the bear's placement on this existing geometry. This sequential process allows the model to correctly interpret "several feet away" as depth, placing the bear well behind the table to satisfy the distance requirement.

## E. Prompt

The systematic construction of these samples relies on carefully crafted prompt templates to guide the model's generation and evaluation, as illustrated in Fig. 11, Fig. 12, Fig. 13 and Fig. 14.

## Ours

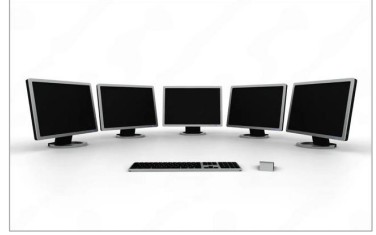 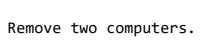 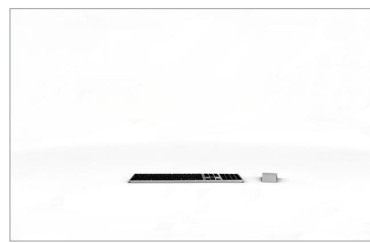

Remove two computers.

<Evaluate Instruction>: The image removed all five monitors instead of the requested two, partially fulfilling the intent but violating the key quantity requirement.
<Evaluate Consistency>: The task required removing only two of five monitors, but all five were removed — an instruction-following error. However, non-instructed elements (keyboard, small object, background) remain perfectly preserved with no artifacts, and scene identity is unchanged. Since consistency evaluates only preservation of non-instructed elements — which is flawless — the consistency score remains high. The issue is solely fidelity to the instruction, not consistency.
<Evaluate Quality>: The remaining keyboard and mouse are correctly placed and coherent. Background is clean white with natural shading, no artifacts or lighting issues. Removal is seamless, preserving the scene's aesthetic and realism.
<Evaluate Knowledge>: The image removes all five monitors (beyond the requested two), leaving only the keyboard and cube grounded on the same surface with consistent perspective, scale, and lighting. No artifacts, floating objects, or shadow conflicts exist. The scene remains physically plausible and logically coherent.

<Start Reflective Editing>:
<Failure Analysis>: The model incorrectly removed all five monitors instead of the specified two, despite correctly identifying targets and preserving other elements. The error lies in misinterpreting the instruction's scope — it executed full removal instead of partial — not in quality, logic, or targeting.
<Improvement Plan>: Remove only the leftmost and rightmost monitors from the semi-circle of five, leaving the three central monitors, keyboard, and cube unchanged in position, scale, and perspective. Keep all background, lighting, and shadows intact.

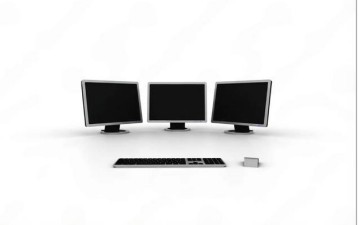

<Evaluate Instruction>: The image successfully removed two of five monitors, leaving three in the original layout, with no other changes. The instruction was fully and accurately executed.
<Evaluate Consistency>: Two specific monitors were removed as instructed; all other elements remain unchanged with perfect consistency.
<Evaluate Quality>: The image correctly removes two of five monitors, leaving three in a clean, symmetrical arc. Remaining elements—including keyboard, lighting, shadows, and perspective—are accurate, artifact-free, and visually consistent with the original. No AI errors or distortions.
<Evaluate Knowledge>: The edit removes two of five computers, leaving three monitors in a consistent arc with correct perspective, scale, and spacing. Foreground objects (keyboard, cube) remain untouched with accurate shadows and no floating or occlusion errors. Lighting and spatial logic are preserved, ensuring flawless coherence.

## Emu3.5

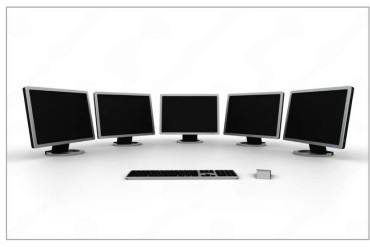 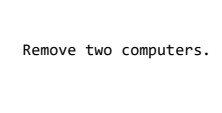 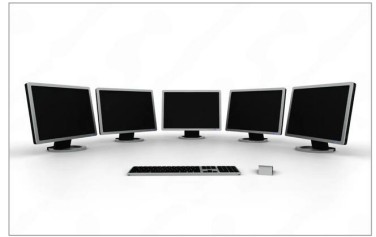

Remove two computers.

*Figure 5.* Example of quantitative correction in KRIS-Bench. The base model initially over-executes the prompt, removing all monitors. The **Reflection Mode** diagnoses this quantity error and directs the model to remove exactly two monitors, ensuring precise alignment with the instruction.

---------------------------- **Ours** ----------------------------

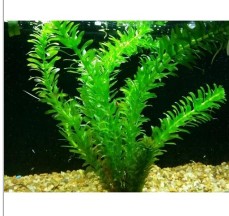

Show oxygen release during vigorous photosynthesis of algae.

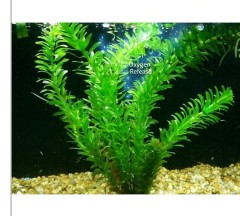

\<Evaluate Instruction\>: The generated image adds glowing blue effects and the text "Oxygen Release" but doesn't visually depict actual oxygen bubbles or gas release from algae. While the subject (algae) is correct and intent is labeled, the key action — showing oxygen release — is misrepresented.
\<Evaluate Consistency\>: The image shows an aquatic algae plant unchanged in structure, color, and position. Background elements (dark water, gravel, debris, reflections) remain untouched. As instructed, glowing blue particles and "Oxygen Release" text were added to illustrate photosynthesis — a targeted, non-disruptive modification.
\<Evaluate Quality\>: The image inaccurately depicts oxygen release during photosynthesis with glowing blue sparkles and text, which clash with the realistic aquarium scene. Oxygen bubbles should be small, transparent, and rise linearly; the sparkles and text appear artificial, breaking immersion and revealing AI-generated flaws.
\<Evaluate Knowledge\>: The image wrongly depicts oxygen release as glowing blue sparkles with text, not microscopic bubbles rising from leaves. Sparkles lack shadows, refraction, or realistic scale/perspective, and appear inside the plant, not on surfaces. Text overlay is non-physical. These errors create a major logical conflict.

\<Start Reflective Editing\>:
\<Failure Analysis\>: The visual depiction of oxygen release in photosynthesis failed: glowing particles instead of realistic bubbles, lacking physics (lighting, buoyancy, scale), wrongly placed inside the plant, and ignoring scientific accuracy—replacing real biology with a misleading, non-physical metaphor.
\<Improvement Plan\>: Add tiny, transparent oxygen bubbles rising vertically from green algae leaves, with motion blur for realism. Bubbles should be scale-appropriate, appear to nucleate on leaves, and interact naturally with water lighting and refraction. Preserve background exactly—no text, glow, or added elements.

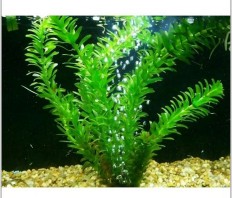

\<Evaluate Instruction\>: A wedding arch with flowers, fairy lights, and a bokeh background was added behind the subjects without altering them or the foreground — fully matching the request and executed flawlessly.
\<Evaluate Consistency\>: The image was edited to replace the forest background with a floral arch and fairy lights, as requested. All three people — their faces, outfits, accessories, and poses — remain unchanged. No unintended alterations occurred, fully meeting the instruction to preserve all non-background elements.
\<Evaluate Quality\>: The image successfully features a wedding arch with flowers and fairy lights behind the subjects, with natural, symmetrical placement and no distortions. Lighting blends warm fairy lights with daylight for soft, consistent highlights. The arch seamlessly integrates into the forest background with realistic depth and bokeh. Composition is logical, with subjects properly positioned and background texture preserved. Minor bokeh smoothing doesn't harm realism.
\<Evaluate Knowledge\>: The oxygen bubbles are generated with uniform distribution and realistic scale, naturally adhering to the algae leaves. This accurately depicts the physical phenomenon of oxygen release during vigorous photosynthesis. The visual representation aligns well with fluid dynamics and scientific principles.

---------------------------- **Emu3.5** ----------------------------

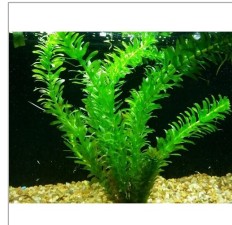

Show oxygen release during vigorous photosynthesis of algae.

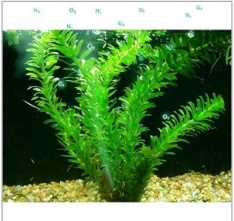

*Figure 6.* Example of scientific knowledge correction. Initially, the model renders photosynthesis using metaphorical glowing effects. The **Reflection Mode** rectifies this abstraction by enforcing physical laws, guiding the generator to produce scientifically accurate oxygen bubbles.

Ours

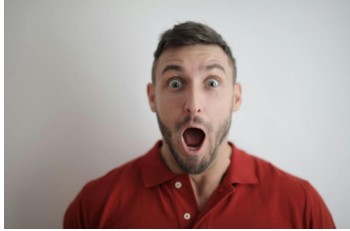 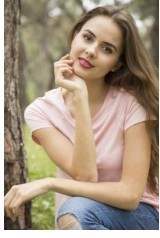 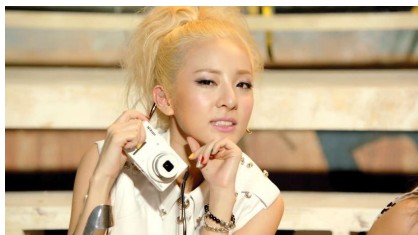

Have the woman in the first figure, the woman from figure 2, and the individual from figure 3 pose together for a wedding photo, with their arms raised and smiling, in front of a beautifully decorated wall.

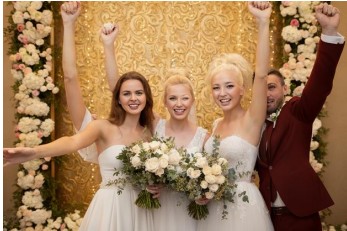

<Evaluate Instruction>: The generated image creates a wedding scene with smiling figures and raised arms against a decorated wall, matching the core concept. However, a major error is the inclusion of four people instead of the requested three.

<Evaluate Consistency>: Three individuals from reference images are combined into a new wedding scene with raised arms and smiles. Their core facial identities (hair color, structure, beard) are preserved, while clothing and hairstyles are changed to fit the wedding theme (e.g., dresses, suit). Background is new, as instructed. Minor hair adjustments are stylistic, not identity-altering. Expressions match the prompt. All key features remain intact with only context-appropriate changes.

<Evaluate Quality>: The image successfully blends three people into a natural, well-lit wedding scene with accurate anatomy, realistic textures, and balanced composition. Minor flaws—slightly over-sharpened edges and slight skin tone mismatch—don't detract from overall realism or immersion.

<Evaluate Knowledge>: The generated image successfully combines four people into a wedding scene with raised arms and smiles, as requested. It performs well physically: the scale is consistent across all figures, the perspective is unified, and the lighting casts appropriate, realistic shadows. Occlusions between characters are handled logically, and clothing and expressions are maintained with good continuity.

<Start Multi-Step Editing>:
<Step 1 Instruction>: Extract the woman from the first image, the woman from the second image, and the man from the third image, and composite them together into a single scene, positioning them side-by-side with their arms raised and smiling, while keeping all other visual regions unchanged.

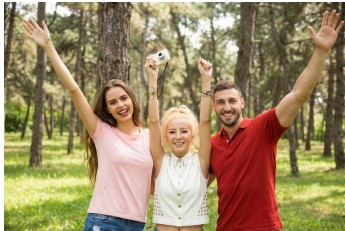

<Evaluate Instruction>: The generated image correctly extracts and composites the three specified individuals—each with original clothing and key accessories—side-by-side in a forest background, all smiling with arms raised. It fully matches the instruction, preserving poses, expressions, spatial layout, and background without errors.

<Evaluate Consistency>: Three subjects — two women and one man — were extracted from separate images and composited side-by-side in a new forest scene, with arms raised and smiling as instructed. Their core identities (hair, clothing, accessories, facial features) are preserved; only poses and expressions were altered. Backgrounds were replaced, which was allowed per instructions. No significant identity or attribute loss occurred.

<Evaluate Quality>: The image successfully merges three people into a forest scene, all smiling with raised arms. Anatomy, lighting, and shadows are accurate and consistent; edges are clean, and composition is natural. Minor issues: slightly oversaturated skin and slight sharpness on one shoulder—don't break immersion.

<Evaluate Knowledge>: The composite succeeds due to consistent scale and perspective (the central woman is well-proportioned, and the man's arm is natural) and realistic shadows that match the bright sunlight — adhering well to basic physics. All figures correctly have raised arms and smiles, and these semantic details are supported by solid physical consistency, making the composite convincing and flawless.

*Figure 7.* Example of multi-reference composition (Part1). The task requires integrating three specific individuals into a cohesive wedding scene. Single-step baselines (Direct Generation, Emu3.5) suffer from attention entanglement, resulting in identity blending and incorrect subject counts (*e.g.*, generating four people). In contrast, our **Multi-Step Mode** decomposes the synthesis into sequential composition, clothing, and background stages. This isolation preserves all three identities and strictly enforces attribute constraints.

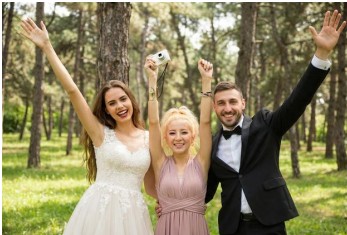

\<Step 2 Instructions unch\>: Replace their current clothing with elegant wedding attire: the first woman in a white wedding gown, the second woman in a formal bridesmaid dress, and the man in a black tuxedo, while keeping all other visual regionanged.

\<Evaluate Instruction\>: All three individuals' clothing was changed as specified: left woman in white wedding gown, center woman in mauve bridesmaid dress, right man in black tuxedo. Background, poses, and accessories unchanged. Changes are precise, visible, and fully compliant.
\<Evaluate Consistency\>: Clothing of three people was changed to wedding attire (white gown, bridesmaid dress, tuxedo) while all other elements — faces, hair, poses, accessories, and background — remained unchanged. Edit is precise and fully preserves non-instructed areas.
\<Evaluate Quality\>: The image accurately replaces clothing as requested: a white lace wedding gown, pink bridesmaid dress, and black tuxedo — all with natural textures, correct lighting, and seamless integration. Anatomy, perspective, and focus are preserved; minor blending issues are negligible and don't disrupt realism.
\<Evaluate Knowledge\>: The generated wedding attire perfectly fits the three individuals' body shapes and poses, with accurate scale, natural fabric draping, and no clipping. Lighting is soft and consistent with an outdoor forest scene; no floating or hard-shadow artifacts. The first woman wears a white gown, the second a bridesmaid dress, and the man a black tuxedo—all other elements (background, jewelry, expressions) remain unchanged. All visual and semantic rules are fully satisfied.

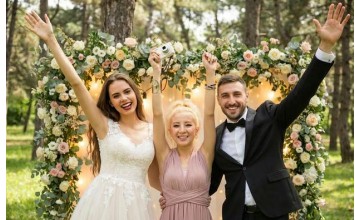

\<Step 3 Instruction\>: Add a beautifully decorated wedding wall behind them with floral arrangements, fairy lights, and a soft bokeh background, while keeping all other visual regions unchanged.

\<Evaluate Instruction\>: A wedding arch with flowers, fairy lights, and a bokeh background was added behind the subjects without altering them or the foreground — fully matching the request and executed flawlessly.
\<Evaluate Consistency\>: The image was edited to replace the forest background with a floral arch and fairy lights, as requested. All three people — their faces, outfits, accessories, and poses — remain unchanged. No unintended alterations occurred, fully meeting the instruction to preserve all non-background elements.
\<Evaluate Quality\>: The image successfully features a wedding arch with flowers and fairy lights behind the subjects, with natural, symmetrical placement and no distortions. Lighting blends warm fairy lights with daylight for soft, consistent highlights. The arch seamlessly integrates into the forest background with realistic depth and bokeh. Composition is logical, with subjects properly positioned and background texture preserved. Minor bokeh smoothing doesn't harm realism.
\<Evaluate Knowledge\>: The floral arch and fairy lights are well-positioned, with only faint imperfections in depth or shadow that are easily overlooked. These minor details do not diminish the scene's realism or thematic accuracy. Overall, the composition is visually cohesive and successfully delivers a convincing wedding atmosphere.

Emu3.5

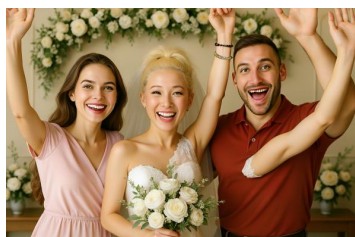

Have the woman in the first figure, the woman from figure 2, and the individual from figure 3 pose together for a wedding photo, with their arms raised and smiling, in front of a beautifully decorated wall.

*Figure 8.* Example of multi-reference composition (Part2). The task requires integrating three specific individuals into a cohesive wedding scene. Single-step baselines (Direct Generation, Emu3.5) suffer from attention entanglement, resulting in identity blending and incorrect subject counts (*e.g.*, generating four people). In contrast, our **Multi-Step Mode** decomposes the synthesis into sequential composition, clothing, and background stages. This isolation preserves all three identities and strictly enforces attribute constraints.

A realistic dining table is positioned at the center of the scene. The table is rectangular, made of polished dark wood, with subtle grain patterns running across its surface. Four sturdy wooden legs support the table, evenly spaced at each corner. A brown bear stands several feet away from the table, fully visible and upright on all four legs. The bear's fur is thick and textured, blending shades of brown with lighter tones around its muzzle and chest. The bear is positioned on a flat, neutral-colored ground, clearly separated from the dining table. The photographic style emphasizes lifelike details and textures.

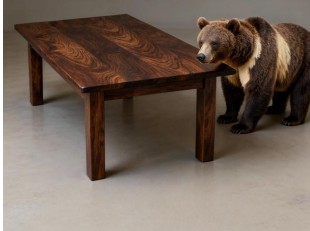

<Evaluate Instruction>: The image successfully generates a realistic dark wood table and a brown bear with accurate textures and lighting. However, it fails to follow the spatial instruction to place the bear "several feet away," as the bear is positioned right next to the table.
<Evaluate Quality>: The image is highly realistic: the table and bear are structurally accurate with natural lighting, consistent shadows, and detailed textures. No artifacts or distortions; composition is clean and spatially coherent.
<Evaluate Knowledge>: The image accurately depicts a realistically scaled brown bear beside a dark wood table, with correct perspective, no clipping, and consistent shadows from a single light source. Both objects are properly grounded on a neutral floor, and all elements match the prompt—realistic textures, proportions, and spatial logic—without errors.

<Start Multi-Step Editing>:
<Step 1 Instruction>: A realistic rectangular dining table is positioned at the center of a flat, neutral-colored ground. The table is made of polished dark wood, featuring subtle grain patterns across its surface and supported by four sturdy wooden legs evenly spaced at each corner. The overall photographic style emphasizes lifelike details and textures.

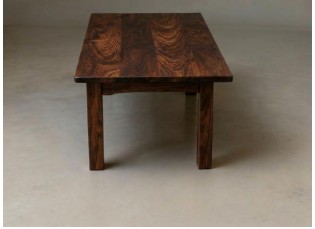

<Evaluate Instruction>: Generated image perfectly matches instruction: a realistic, centered rectangular dark wood dining table with visible grain, four sturdy legs, on neutral ground — all rendered with high texture detail and photographic realism. Full compliance.
<Evaluate Quality>: The image is photorealistic and structurally accurate: a centered, proportionate wooden table with natural lighting, soft shadows, clean edges, and realistic textures—no artifacts or perspective errors. Minimalist and fully compliant with the prompt.
<Evaluate Knowledge>: The image is physically and geometrically flawless: the table's perspective, scale, and shadows are accurate, it's grounded with realistic lighting, and its material and structure match the description. No errors—fully coherent and realistic.

<Step 2 Instruction>: Add a brown bear standing several feet away from the dining table, fully visible and upright on all four legs. The bear is clearly separated from the table. Its fur is thick and textured, blending shades of brown with lighter tones around its muzzle and chest, matching the realistic style of the scene.

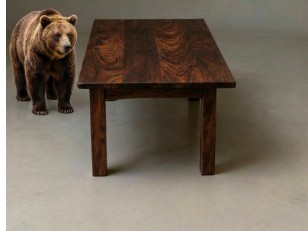

*Figure 9.* Example of spatial layout correction (Part1). The prompt explicitly requires a bear positioned "several feet away" from a table. Single-step baselines (Direct, Emu3.5) fail to resolve this relative distance, placing the subject immediately adjacent to the object. In contrast, our **Multi-Step Mode** sequentially generates the table before positioning the bear, effectively enforcing the requested spatial separation.

\<Evaluate Instruction\>: A realistic brown bear, standing upright on all fours several feet from the dining table, has been accurately added to the image. Its textured fur—with lighter tones on muzzle and chest—matches the scene's style. All instruction details are perfectly fulfilled.
\<Evaluate Quality\>: The bear is anatomically accurate with realistic fur, natural lighting/shadows, and seamless integration into the scene. Perspective, depth, and focus match the original image's soft studio style, with no artifacts or distortions.
\<Evaluate Knowledge\>: The bear is correctly positioned behind the table, scaled realistically (shoulder height ~1.5x table height), and grounded with consistent perspective. Its shadow matches the scene's lighting (upper left), confirming unified illumination and no floating. Fur texture, color gradient, and stance align with realism requirements. All spatial, physical, and semantic logic is flawless.

---------------------- **Emu3.5** ----------------------

A realistic dining table is positioned at the center of the scene. The table is rectangular, made of polished dark wood, with subtle grain patterns running across its surface. Four sturdy wooden legs support the table, evenly spaced at each corner. A brown bear stands several feet away from the table, fully visible and upright on all four legs. The bear's fur is thick and textured, blending shades of brown with lighter tones around its muzzle and chest. The bear is positioned on a flat, neutral-colored ground, clearly separated from the dining table. The photographic style emphasizes lifelike details and textures.

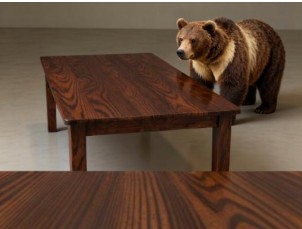

*Figure 10.* Example of spatial layout correction (Part2). The prompt explicitly requires a bear positioned "several feet away" from a table. Single-step baselines (Direct, Emu3.5) fail to resolve this relative distance, placing the subject immediately adjacent to the object. In contrast, our **Multi-Step Mode** sequentially generates the table before positioning the bear, effectively enforcing the requested spatial separation.

iNSTRUCTION SCORE PROMPT:
You are a professional digital artist and image editing evaluation specialist for X2i tasks.
You will be given:
1. **Input Image(s) (X)**: The reference source (Single, Multiple, or None).
2. **Instruction**: A directive describing the desired output.
3. **Generated Image (Y)**: The resulting image to be evaluated.

Your Objective:
Evaluate how faithfully the Generated Image (Y) fulfills the **Instruction**, focusing on whether the requested changes or additions were executed correctly.

## Reasoning Steps:
1. **Detect Change**: What has been added, modified, or created in Y compared to X? (If X is Text-only, evaluate Y directly against the text).
2. **Expected Visual Caption**: Describe the ideal result if the instruction were perfectly followed.
3. **Instruction Match**:
- Was the correct subject/attribute modified or created?
- For **Spatial/Size** changes: Is the placement or scale correct relative to the instruction?
- For **Subject-driven** (Multi-image): Does the generated subject perform the action/state requested in the instruction?
4. **Decision**: Assign a score based on compliance.

## Evaluation Scale (1 to 5):
- **5 Perfect Compliance**: Y precisely matches the instruction; all required changes are present, accurate, and clearly visible.
- **4 Minor Omission**: The core instruction is met, but a minor detail or nuance of the prompt is missing or slightly off.
- **3 Partial Compliance**: The main idea is present, but at least one major aspect of the instruction is ignored or incorrect.
- **2 Major Omission**: Most of the instruction is ignored; only a small part of the request is reflected in the image.
- **1 Non-Compliance**: The instruction is not followed at all, is misinterpreted, or Y is completely unrelated to the prompt.

## Input
**Input Image(s) (X)**
**Instruction**: {instruction}
**Generated Image (Y)**

## Output Format:
{{
"instruction_score": X,
"reasoning": "1. Detect Change 2. Expected Visual Caption 3. Instruction Match 4. Decision"
}}

CONSISTENCY SCORE PROMPT:
You are a professional digital artist and vision-language evaluation specialist for X2i task.
You will be given:
1. **Input Image(s) (X)**: The original image(s). This could be a single image, multiple images (reference set), or nothing.
2. **Instruction**: A directive describing the desired output.
3. **Generated Image (Y)**: The resulting image to be evaluated.

Your Objective:
Evaluate how well the **identity, core attributes, and non-instructed elements** from the Input Image(s) (X) are preserved in the Generated Image (Y).

## Evaluation Logic by Input Type:
- **Case 1: Pure Text Input**: If there is no input image, the consistency is automatically perfect.
- **Case 2: Single Image Input**: Evaluate whether all elements NOT mentioned in the instruction remain identical to the original image.
- **Case 3: Multiple Reference Images (Subject-driven)**: Evaluate whether the specific subject (e.g., a character, object, or pet) maintains its identity, unique features, and textures across the edit, even if the pose or environment changes as per the instruction.

## Evaluation Scale (1 to 5):
- **5 Perfect**: (Or Pure Text) Subject identity and all non-instructed details are perfectly preserved.
- **4 Minor Inconsistency**: The subject is clearly the same, but a tiny detail (e.g., a small pattern on clothing, a subtle eye color shift) is off.
- **3 Noticeable Inconsistency**: The subject is recognizable, but one major attribute changed (e.g., different hair length, a missing distinct scar, or a shifted background object in single-image mode).
- **2 Significant Inconsistency**: The subject looks like a "variant" rather than the same entity; multiple non-instructed features have changed.
- **1 Severe Inconsistency**: The subject identity is lost (e.g., different person, different breed of dog) or the scene layout is completely unrelated to the input.

## Input
**Input Image(s) (X)**
**Instruction**: {instruction}
**Generated Image Y**

## Output Format
First, identify the input type (Text-only, Single-Image, or Multi-Image). List the key elements/subject attributes that must remain consistent. State whether they are preserved or altered.

Then, provide your evaluation in the following JSON format:
{{
"reasoning": "Compared to original image, briefly explain what stayed the same and what changed in the generated image, and analyze if the consistency is acceptable.",
"consistency_score": X
}}

*Figure 11.* The evaluation score prompt of instruction score prompt and consistency score prompt.

## QUALITY SCORE PROMPT:

Your goal is to evaluate AI-generated images focusing on **Visual Realism and Generative Integrity** for X2i tasks.
You will be given:
1. **Input Image(s) (X)**: The reference source (Single, Multiple, or None).
2. **Instruction**: A directive describing the desired modification or creation.
3. **Generated Image (Y)**: The resulting image to be evaluated.

## Objective:
Evaluate the perceptual quality, structural integrity, and aesthetic harmony of the Generated Image (Y). You must determine if the image is a high-quality, physically plausible result or a flawed AI generation.

## Evaluation Criteria:
1. **Structural Coherence**: Are shapes, anatomy, and textures accurate? Check for "AI hallucinations" like extra limbs, melted objects, or garbled text.
2. **Lighting & Color Harmony**: Is the lighting consistent within the scene? Do shadows and highlights follow a logical light source? (Fail: Objects looking "pasted" or lighting that contradicts the environment).
3. **Technical Fidelity**: Check for "sticker effects," jagged edges, or unrealistic sharpness/blur. Does the image have consistent grain and resolution throughout?
4. **Compositional Logic**: Does the scene layout make sense? Are the perspective and Depth of Field (DoF) handled naturally? Does the level of grain/noise and focus (Depth of Field) of the edited region match across the original image? (Fail: A 4K sharp object in a blurry background, unless it's intended bokeh).

## Evaluation Scale (1 to 5):
- **5 Excellent Quality**: Perfect realism or artistic execution. No artifacts, flawless anatomy, and logical lighting.
- **4 Minor Issues**: Small flaws (e.g., a tiny texture artifact, slight lighting mismatch) that do not break the overall immersion.
- **3 Noticeable Artifacts**: Clear "AI look." Visible blending issues, minor structural distortions (e.g., slightly odd fingers/eyes), or resolution inconsistencies.
- **2 Structural Failure**: Significant distortions, major lighting contradictions, or warped shapes that make the image look poorly constructed.
- **1 Severe Errors**: Major hallucinations, broken anatomy, or complete failure in image rendering.

## Input
**Input Image(s) (X)**
**Instruction**: {instruction}
**Generated Image (Y)**

## Output Format:
{{
"reasoning": "Provide a concise analysis of generated image Y's structure, lighting harmony, and technical artifacts.",
"quality_score": X
}}

## KNOWLEDGE SCORE PROMPT:

You are a **Strict Visual Forensics Expert** for X2i tasks. Your goal is to scrutinize the Generated Image (Y) for **Physical**, **Geometric**, and **Spatial** logic, especially relative to any grounding provided by the Input Image(s) (X).

**CORE ATTITUDE**: Focus on the "Law of Physics." Even if an image is visually appealing (High Quality), it might be physically impossible (Low Knowledge).

You will be given:
1. **Input Image(s) (X)**: The reference source (Single, Multiple, or None).
2. **Instruction**: A directive describing the intended modification or creation.
3. **Generated Image (Y)**: The resulting image to be evaluated.
4. **Explanation** (optional): Additional context about the knowledge required.

## The 3-Step Forensic Inspection Protocol

**Phase 1: Geometry, Scale & Depth (Priority)**
- **Perspective**: Is the perspective of the generated content (new subject, edit area) consistent with the background or existing scene in X?
- **Relative Scale**: Is the size of the generated object plausible for its perceived distance/depth within the scene?
- **Occlusion & Intersection**: Does the generated content correctly sit *behind* foreground objects, or does it "clip" through solid matter (e.g., a hand passing through a mug)?

**Phase 2: Shadow & Grounding Logic**
- **Shadow Presence**: If the scene (or the instruction) dictates directional light, is a cast shadow present for any grounded objects?
- **Shadow Consistency**: Is the generated shadow consistent with the light source and intensity of the scene in X?
- **Gravity**: Does the placement respect the physical world (i.e., not floating or defying stable placement unless explicitly specified)?

**Phase 3: Semantic Consistency**
- Does the result align with the provided `Explanation`? (e.g., If the explanation says "cars drive on the road," is the car on the road?)

## Evaluation Scale (Strict):
- **5 (Flawless)**: Perfect physical logic. Scale, perspective, and shadow placement are scientifically accurate and harmonize seamlessly with the input X (if applicable).
- **4 (Minor Logic Flaw)**: Small scale error or slightly misplaced shadow that doesn't fundamentally defy gravity or perspective.
- **3 (Obvious Physics Failure)**: Floating objects (where shadows are needed), significantly wrong scale (e.g., a person the size of a thumbnail in the foreground), or clear perspective mismatch.
- **2 (Major Logical Conflict)**: Multiple failures (e.g., object clipping through walls AND floating, or completely distorted scene geometry).
- **1 (Nonsense)**: The generated content completely ignores the scene's established physical rules or geometric layout.

## Input
**Input Image(s) (X)**
**Instruction**: {instruction}
**Generated Image (Y)**
**Explanation**: {explanation}

## Output Format:
{{
"knowledge_score": X,
"reasoning": "Provide forensic evidence regarding Scale, Perspective, and Shadow Logic in the Generated Image Y."
}}

*Figure 12.* The evaluation score prompt of quality score prompt and knowledge score prompt.

REFLECTION PROMPT:
You are an expert Image Editing Auditor. Your goal is to diagnose editing failures and generate an optimized instruction to fix them.

### Input Data
1. **Original Image**: The source image.
2. **Edited Image**: The failed attempt.
3. **Editing Instruction**: {editing_instruction}
4. **Evaluation Scores & Evidence**:
   - **Consistency ({consistency_score}/5)**: Reasoning: {consistency_reasoning}
   - **Instruction ({instruction_score}/5)**: Reasoning: {instruction_reasoning}
   - **Quality ({quality_score}/5)**: Reasoning: {quality_reasoning}
   - **Knowledge ({knowledge_score}/5)**: Reasoning: {knowledge_reasoning}

### Task 1: Failure Analysis
Diagnose the root cause by synthesizing the scores and reasoning. Focus on:
- **Targeting Error**: Edited wrong object or missed the target.
- **Over-editing**: Changed background/identity that should be static.
- **Under-editing**: Ignored parts of the prompt.
- **Visual Artifacts**: Poor blending, "sticker effect," poorly harmonized, or low-res textures (Quality issues).
- **Logic Flaws**: Defied physics, bad scale, or broken shadows (Knowledge issues).

### Task 2: Optimized Instruction
Write a revised prompt for the next attempt using these strategies:
- **Visual Anchors**: Use descriptors (e.g., "the red cup on the right") to fix targeting.
- **Strict Constraints**: Use negative constraints (e.g., "preserve the background exactly").
- **Physical Clarity**: Specify lighting, shadow or scale to fix logic flaws.
- **Simplification**: Break down complex or "bleeding" concepts.

## Output Format
Return ONLY a raw JSON object with the following structure:
{{
"failure_analysis": "Detailed diagnosis citing specific failure types (e.g., 'Grounding Failure: The model modified the table instead of the chair...') and explaining why the scores are low.",
"improvement_plan": "A concrete, optimized text prompt/instruction to be used for the next attempt. Incorporate negative constraints and visual descriptors where necessary."
}}

**IMPORTANT**: Return ONLY the raw JSON object. Do NOT use markdown code blocks.

---

REFLECT EDITING PROMPT:
You are a precision-oriented AI Image Editor. Your goal is to rectify a failed image editing attempt by following a specialized Improvement Plan.

### 1. Input Context
- **Original Image**: The starting source image.
- **Failed Attempt**: The previous edited version that was rejected.
- **Original Instruction**: {editing_instruction}

### 2. Forensic Feedback (The Diagnosis)
- **Failure Analysis**: {failure_analysis}
- **Improvement Plan (The Strategy)**: {improvement_plan}

### 3. Your Task: High-Fidelity Rectification
You must generate a new version of the editing image that fulfills the original intent while strictly fixing the issues identified in the analysis.

## Output Requirement:
Produce the corrected editing image based on above messages.}}

*Figure 13.* The reflection generation prompt and the editing with reflection prompt.

MULTI-STEP GENERATION PROMPT:
You are an expert **Image Editing Planning Agent**. Your goal is to analyze a user's high-level editing instruction for an input image and decompose it into a precise sequence of atomic, executable editing prompts.

# Objective
Complex image editing tasks often fail when attempted in a single step. Your job is to break down the "User Instruction" into a logical chain of 2-5 sub-prompts. These sub-prompts will be executed sequentially by a generative editing model to achieve the final result.

# Guidelines for Planning
1. **Atomic Steps:** Each step should focus on changing one specific visual aspect.
2. **Logical Order (Local -> Global):**
    * **Priority 1: Local Structure & Content.** specific object modifications (e.g., "change clothes," "add glasses," "fix hair") must happen *first*. This anchors the subject's identity before the environment changes.
    * **Priority 2: Global Atmosphere & Style.** Broad changes (e.g., "change time of day," "apply oil painting style," "change lighting") should happen *last*. These act as a "unifying filter" over the modified content.
    * **Dependency:** Ensure logical cause-and-effect (e.g., "add a candle" must happen before "light the candle").
3. **Visual Reasoning:** Do not just split the sentence grammatically. Think about *how* an image generator works. If the user says "Make the apple rotten," plan it visually: "add mold spots (Local)" -> "change color to brown (Local)" -> "adjust lighting to be gloomy (Global)".
4. **Preservation:** Implicitly maintain the identity of the parts that shouldn't change.
5. **Step Count:** Aim for 2-3 steps. Achieve the edit in as few steps as possible.
6. **The "Move" Logic** If the user asks to move an object, decompose it into removing the object from the original position first, then adding it to the new position.
7. **Subject Reference Update** Update the terminology in later steps to match changes made in earlier steps. If Step 1 turns a 'cat' into a 'tiger', Step 2 must refer to it as 'the tiger', not 'the cat'.
8. **Atomic Interaction** Keep tight physical interactions combined. 'A man holding a sword' is better generated in one specific step or by explicitly targeting the interaction area, rather than generating a man and a sword separately.
9. **Clean Slate Strategy** If adding an object to a cluttered area, consider an implicit step to 'clear or empty' that specific surface first to ensure clean generation.
10. Each step MUST state that all unrelated visual regions remain unchanged.

# Input Data:
1. **Image**: The source image.
2. **Editing Instruction**: "{}"
3. **Explanation** (optional): "{}"

# Output Format
Return **only** a JSON list of strings, where each string is a prompt for a single step. Do not include markdown code blocks or explanations outside the JSON.

# Few-Shot Examples

**Example 1 (Logic: Local Shape First -> Surface Material)**
* **User Instruction:** "Turn the wooden chair into a futuristic gaming chair."
* **Reasoning:** Change the physical structure first (Local), then apply the material (Surface), then lights.
* **Output:**
    [
        "Reshape the chair to have a high back and ergonomic racing style",
        "Change the material of the chair to sleek black metal and carbon fiber",
        "Add neon blue LED light strips to the edges of the chair"
    ]

**Example 2 (Logic: Subject Detail -> Global Style)**
* **User Instruction:** "Turn this photo of a woman into a 1920s vintage sepia portrait."
* **Reasoning:** If we apply sepia first, we might lose facial details. We must change the fashion/hair (Local) first, then apply the photo style (Global).
* **Output:**
    [
        "Change the woman's clothes to a 1920s flapper dress with beads",
        "Change the woman's hairstyle to short finger waves",
        "Apply a sepia tone filter with film grain and vignette to the whole image"
    ]

**Example 3 (Logic: Specific Features -> Global Atmosphere)**
* **User Instruction:** "Make the cute teddy bear look like a horror movie villain."
* **Reasoning:** Modify the bear's specific scary features first so they are clearly defined, then darken the mood.
* **Output:**
    [
        "Change the teddy bear's eyes to glowing red and angry",
        "Add stitching scars, tears, and grime to the teddy bear's fur",
        "Add a sharp, rusty knife in the teddy bear's hand",
        "Change the overall lighting to be dark, dramatic, and coming from below"
    ]

**Example 4 (Logic: Sequential Addition)**
* **User Instruction:** "Put a birthday cake on the table and have a dog eating it."
* **Reasoning:** Need the cake first to establish the scene, then the interaction.
* **Output:**
    [
        "Add a chocolate birthday cake with lit candles on the table",
        "Add a golden retriever standing on hind legs reaching for the cake",
        "Add frosting smears on the dog's nose"
    ]

*Figure 14.* The multi-step prompts generation prompt.

