# OpenReview forum: "Breaking Dual Bottlenecks: Evolving Unified Multimodal Models into Self-Adaptive Interleaved Visual Reasoners"
_ICML.cc/2026/Conference — ICML 2026 regular_

### Official Review · Reviewer_RbLL · 2026-03-02

**Soundness:** 3
**Presentation:** 3
**Significance:** 3
**Originality:** 3
**Overall Recommendation:** 4
**Confidence:** 4

**Summary:**

This paper aims to solve complicated X2I queries. Based on a fundamental ANALYSER-GENERATOR framework, the authors propose three generation paradigms, including direct generation, self-reflection, and multi-step generation. Utilizing frontier VLMs as the ANALYSER and image generative models as the GENERATOR, the authors first curate 50K high-quality examples for SFT and further design a GRPO-based RL pipeline to further enhance the model capability. Extensive experiments demonstrate the effectiveness of the proposed method.

**Compliance With Llm Reviewing Policy:**

Affirmed.

**Key Questions For Authors:**

Check Weakness for details.

**Limitations:**

Overall, this is solid work with clear motivation, well-collected datasets, and extensive experiments throughout SFT and RL. Although not many technical novelties, the engineering efforts are extensive to me. My main concerns lie in the scalability of this method, since currently, it seems to utilize carefully-designed patterns to achieve better benchmark performance instead of general reasoning capability.

**Strengths And Weaknesses:**

- Strengths:
  - The idea is simple: by curating data with specific patterns to enhance the corresponding abilities.
  - The data curation procedure is detailed.
  - Although straightforward, the efforts conducted in this paper are thorough, from data curation to SFT and RL.

- Weakness:
  - About SFT:
    - It is fantastic to see the three inference paradigms designed by the authors, which, however, are novel patterns introduced by the authors manually during SFT instead of models' intrinsic capabilities.
    - In other words, how can we know that these three paradigms are optimal?
    - More details about the human annotation might be necessary, including annotation guidelines, statistics of annotators, and consistency between annotators.
  - About RL:
    - What is the reward model? Is it also a general VLM as the ANALYSER?
    - How will the outcome-based reward face the risk of reward hacking, since Eqn. 2 consists of four items, all requiring reward modeling?
  - Overall, I do agree that this work provides a valid solution with extensive experiments to improve X2I performance, but whether it is the optimal one remains unclear. But of course, this is a bigger question that might go beyond the scope of this paper.

---

> ### Author Rebuttal · Authors · 2026-03-31
>
> We appreciate the positive evaluation and the reviewer’s helpful feedback. While our approach offers a workable solution, we recognize it as an initial study rather than an ultimate one. Broadly speaking, the unified modeling of comprehension and synthesis remains a core challenge in the field. Our work shows that interleaved reasoning represents a concrete and productive path toward this goal.
>
> ### W1:About SFT
> - We appreciate this observation and agree that these inference paradigms are unlikely to emerge as spontaneous behaviors during initial training stages. Our approach stems from the recognition that acquiring complex structured behaviors through unguided reinforcement learning remains a significant challenge for unified models. Therefore, during the SFT phase, we seed the training data with foundational structures inspired by established problem-solving heuristics like divide-and-conquer and iterative refinement. The objective is not to manually define the final policy but to establish a behavioral scaffold that enables effective subsequent learning. Empirical results demonstrate that incorporating these structured priors via SFT measurably boosts both reasoning accuracy and generation fidelity.
> - Regarding the choice of paradigms, we do not posit that these three modes represent the absolute global optimum for unified models. Given that an exhaustive search for optimal reasoning patterns is computationally prohibitive, we employ a curated set of human-inspired priors to structure the model behavior in a tractable manner. Our focus is on demonstrating the utility of this scaffold as a viable baseline rather than presenting it as a definitive solution. Identifying more sophisticated reasoning paradigms remains a vital objective for future research.
> - Regarding the detailed annotation guidelines, we summaried them as follow:
>
>     ``We utilize a two-stage protocol where two independent annotators must mutually agree to retain a synthesized trajectory. First, annotators evaluate the final output quality across four dimensions: instruction following, consistency preservation, visual quality, and physical plausibility. This evaluation directly corresponds to the defined outcome reward. Second, for samples utilizing intermediate steps, annotators assess reasoning coherence—verifying accurate failure diagnosis in reflection-mode and logical, non-redundant task decomposition in multi-step-mode. This stage explicitly aligns with the step-wise reasoning reward. Finally, samples are strictly rejected for omitted instructions, unintended alterations, rendering artifacts, or illogical reasoning chains.``
>
>     To strictly adhere to ICML rebuttal policies regarding external links, we have currently limited our shared content to tables and figures. Since the full annotation guidelines are text-intensive, we are cautious about whether rendering the complete text as an image might be perceived as a policy violation.
>     If the reviewer confirm that providing a rendered image of the full text is permissible, we are prepared to share the complete guidelines via a link during the second-round discussion phase.
>     **In any case, the full annotation guidelines will be included in the final version of the manuscript.**
> - Regarding annotator statistics, 20 students (undergraduate and master's), aged 18 to 26, were organized into 10 pairs for data annotation.
> The Cohen’s $\kappa$ coefficient between the two annotators within each group is 0.86 on average.
> We will include a summary of these details in the revised manuscript.
>
> ### W2:About RL
> - The reward model for both outcome reward and step-wise reasoning reward utilizes an LMM-based approach, *i.e.*, the ANALYZER in our submission, with prompts as detailed in **Appendix E**, where the outcome reward aligns with standard protocols in recent generation literature \[1-3\] and the step-wise reasoning reward is introduce in our framework and proven to be useful to improve the generation quality in Table 3 of origainl submission.
> - We observe no evidence of reward hacking; specifically, the four outcome sub-rewards increased concurrently without displaying a pattern where one metric improves at the expense of others. This stability stems from the fact that the four dimensions are coupled and tied to the same final image, making them difficult to exploit individually. For instance, gains in instruction following that compromise consistency or quality generally fail to raise the total evaluation score.
>
> \[1\] KRIS-Bench: Benchmarking Next-Level Intelligent Image Editing Models.
>
> \[2\] WorldEdit: Towards Open-World Image Editing with a Knowledge-Informed Benchmark.
>
> \[3\] WiseEdit: Benchmarking Cognition- and Creativity-Informed Image Editing.

---

> > ### Author Rebuttal · Reviewer_RbLL · 2026-04-04
> >
> > Thank the authors for the detailed rebuttals. Again, from the scope of this paper only, I think this is a quite complete paper with clear motivation, constructed datasets, and well-trained models. However, as also agreed by the authors, whether the proposed three paradigms can become the next mainstream and be followed by others is still an open question.
> >
> > Therefore, I tend to retain my original score as 4: Weak Accept.

---

> > > ### Author Response · Authors · 2026-04-06
> > >
> > > We genuinely appreciate the time and effort you have dedicated to engaging with our work. All clarifications, additional experiments, and qualitative analyses developed during the rebuttal will be incorporated into the final version.

---

### Official Review · Reviewer_fTr6 · 2026-03-09

**Soundness:** 3
**Presentation:** 3
**Significance:** 3
**Originality:** 3
**Overall Recommendation:** 4
**Confidence:** 4

**Summary:**

The paper introduces a multi-step framework that leverages iterative self-reflection and task-decomposition to improve text-based image generation and editing capabilities of unified models. The authors first leveraged stronger model to collect demonstrations that foster such behavior for supervised finetuning, then performed reinforcement learning to further improve performance as well as efficiency. The authors showed the effectiveness of their framework by evaluating their finetuned models on GenEval, KRIS-Bench and OmniContext benchmarks with competitive performance, and the contribution of different designs e.g., multi-step reasoning, RL objectives through controlled ablations.

**Compliance With Llm Reviewing Policy:**

Affirmed.

**Final Justification:**

Authors have addressed most of my concerns in the rebuttal and the follow-up response. While scalability issues remain, the core arguments are much more convincing under the new experiments, so I am raising the soundness score from 2 to 3 and subsequently the overall rating to 4.

**Key Questions For Authors:**

* While truncating loss to only the last round in self-reflection incentivizes the model to generate the correct edit more than having loss over the entire trajectory, having loss over the entire trajectory might better curate the right behavior e.g., iterative error discovery and correction. I wonder if authors have had an ablation study on where to apply the loss on these?
* While the authors applied a structured pipeline for trajectory-level data generation, I wonder where the seed data comes from e.g., the beginning visual and text input for the image editing task for SFT?
* While models achieve competitive performance on image editing, I wonder how models perform on other evaluations e.g., image understanding after such post-training, given the models being applied here are unified models that are intended to do more than text-based image editing?
* In the impact of reasoning modes ablation, I wonder if authors can instead control the number of trained tokens compared to the number of instances, as the two stronger reasoning modes also come with more tokens that models are trained on compared to the direct only baseline.

**Limitations:**

The authors did not discuss potential negative impact and limitations. While I don't see any immediate potential negative impact that the authors' contribution may have, one limitation is the reliance on stronger models for data generation and the involvement of human verification. An open direction would be to eliminate the need for stronger models (i.e., self-evolution) and the need for human verification (i.e., self-supervision).

**Strengths And Weaknesses:**

### Strengths
* The authors' proposed framework, together with the dataset that the authors generated and verified is effective and the finetuned Emu3.5 achieved competitive result on text-to-image generation and editing benchmarks.
* The authors demonstrated that, at the instance level (e.g., first part of the ablation), structured multi-step reasoning is often more effective than direct generation. This steers the community into thinking scaling multi-step reasoning as an axis for better unified models.
* The designed RL objectives are effective, and authors clearly showed how different reward objective contributes to task performance and efficiency.

### Weaknesses
* Several presentation issues that can be improved:
  * L26: "understanding-generation gap" it's not easily to directly understand this phrase (which lags behind in what way) without a specific definition e.g., instruction-following image editing lags behind image understanding.
  * L33: Authors mentioned the attention entanglement and visual refinement bottleneck then traced back their occurrences to a few reasons, but the terms themselves are not concretely defined upfront e.g., what is attention entanglement in the context of X2I?
  * L324: This is rather a presentation issue -- as a highlighted (and the first set of) result, the corresponding table should not be rather referenced late in the appendix.
  * L425: This is another presentation issue, but it appears that in absolute performance SFT+RL without complexity penalty yields the best result, but the bolded items are all on the full SFT+RL part. Despite faithfully mentioning the tradeoff in the paragraph, authors should make the table clear as well.
 * Both the critique e.g., Qwen3VL-235B and the actor e.g., Gemini-3-Pro-Image in the data pipeline are possibly much stronger models in their needed capabilities compared to the models that authors aim to optimize their models with. Therefore, it is unclear whether the improvements come from distilling capabilities from stronger models or the proposed framework that involves direct generation, self-reflection and multi-step planning. To argue for the utility of the framework itself, authors should instead rely on the base model to generate the data for post-training that learns the set of behaviors that fit into the framework. This affects multiple claims in the manuscript that mention the framework as a strength.
 * The dataset is expensive to collect (e.g., by calling frontier commercial models for image generation) and non-trivial to verify (all instances undergo human verification process). Thus training data scalability is poor. A useful study here is to eliminate commercial models (e.g., rely on the same base model as my point above) and human verification to generate training data and report performance.

---

> ### Author Rebuttal · Authors · 2026-03-31
>
> ### W1:Presentation
> - In Lines 12–25, we analyze the **understanding-generation gap**, a phenomenon documented in recent studies[1-2]. We observe that unified models often correctly interpret semantic intent yet fail to execute accurate generation for precise instruction following.
> - Lines 32–40 address the **attention entanglement**[3-4] and **visual refinement**[5] bottlenecks. Attention entanglement occurs when a model fails to decouple multiple objects or relations during single-pass generation, resulting in mutual constraint interference. Visual refinement bottleneck refers to instances where the model captures the global intent, but the output contains local defects requiring structured correction.
> - Due to space constraints, **Table 5** is located in the Appendix and cited in the main text.
> - We appreciate the feedback and will refine them in following version.
>
> [1] An Empirical Study of GPT-4o Image Generation Capabilities
>
> [2] UniCorn: Towards Self-Improving Unified Multimodal Models through Self-Generated Supervision
>
> [3] Fair generation without unfair distortions: Debiasing text-to-image generation with entanglement-free attention.
>
> [4] Translation of text embedding via delta vector to suppress strongly entangled content in text-to-image diffusion models
>
> [5] Omnigen2: Exploration to advanced multimodal generation.
> ### W2:Training mode
> - To determine if improvements stem primarily from stronger-model distillation rather than our framework, we trained the base model on the complete dataset using only direct-generation supervision.
> - Results in **<https://anonymous.4open.science/r/re16701-5BA0/Gen_only.md>** lack comparable improvements, indicating that performance gains are not driven solely by the inclusion of teacher-generated data, but by structured supervision across direct generation, self-reflection, and multi-step planning.
> - We attribute this to the fact that Emu3.5 was pre-trained on extensive high-quality synthetic data; thus, additional teacher outputs alone yield diminishing returns.
> ### W3-Q2:Data source and collection
> - Trajectory construction is initialized with de-identified production data authorized for research, providing initial visual inputs and instructions. To broaden coverage, we include images from public datasets, i.e., UnicEdit-10M and UniREditBench and will clarify these sources in the revised manuscript.
> - Our measurements indicate that the base model's success rate for valid training instances is approximately one-quarter of Gemini’s. Therefore, replacing the base model with Gemini offers no clear cost advantage, even before accounting for differences in output quality. Using the Gemini 3 Pro Image Batch API, the average generation cost is `$0.067` per image. By comparison, Emu3.5 requires an 80GB GPU and averages 40s of compute time per image. Based on Google Colab A100 pricing (`$4.713/hour`), the estimated cost per successful image for the base model is `$0.2095`, **which is 3.13x higher than the Gemini API rate**, excluding overhead for storage, maintenance, and retry scheduling.
> - The human verification step is a deliberate quality control measure. It prevents the propagation of subtle semantic or reasoning errors from automated trajectories into the final training phase.
> ### Q1:Trajectory loss
> - The design choice is evaluated during preliminary experiments. Applying loss to the full trajectory yields no clear benefit for reflection and introduces error-prone patterns from failed outputs. Conversely, restricting loss to the final diagnosis, reflection prompt, and successful correction results in superior performance. We report the corresponding SFT ablation results in **<https://anonymous.4open.science/r/re16701-5BA0/Selective_loss.md>**. These results validate our selective loss masking, prioritizing correction logic while avoiding the imitation of low-quality steps.
> ### Q3:Understanding ability
> - In the original submission, we focused on X2I benchmarks because Emu3.5 is designed as a unified world model rather than a general-purpose VLM. Thus, our approach prioritizes generation-time reasoning and editing fidelity over standard multimodal tasks. To assess whether training impacts general multimodal comprehension, we evaluate the BAGEL-based variant (**Reviewer#E1hH-W1**) on two VLM benchmarks (**<https://anonymous.4open.science/r/re16701-5BA0/BAGEL_VLM.md>**). We observe no significant degradation following our training procedure. This suggests that our approach does not isolate generation; instead, planning and reflection tasks inherently require and reinforce multimodal understanding.
> ### Q4:Training under controlled tokens
> - Using the token budget of the Direct mode, we developed token-matched variants for the other reasoning datasets, with results in **<https://anonymous.4open.science/r/re16701-5BA0/Token_matched.md>**. These findings align with Table 3, indicating that improvements arise from reasoning supervision rather than increased token volume.

---

> > ### Author Rebuttal · Reviewer_fTr6 · 2026-04-03
> >
> > Thanks for the response. In W1, the authors clarify key terms, but my concern remains that the terminology itself is not easily interpretable without additional context; clearer naming would improve accessibility in a revision. W2 does not address my concern. The ablation shows that structured supervision (reflection/planning) outperforms direct-generation supervision, but both settings rely on data generated by strong teacher models. As such, it does not isolate whether gains arise from the proposed framework or from distilling capabilities of stronger models. For W3, the response focuses on cost comparison rather than scalability, which only partially addresses my concern. It does not evaluate whether the pipeline can operate effectively without reliance on commercial models or human verification, which is central to assessing scalability. I appreciate authors responses to my Q1-Q4 -- these are fully addressed!

---

> > > ### Author Response · Authors · 2026-04-06
> > >
> > > ### W1: Presentation
> > >
> > > Thank you for your suggestion. In the revised version, we will add clearer definitions, intuitive explanations, and concrete examples so that readers with different levels of familiarity can understand the concepts more easily.
> > >
> > > ---
> > >
> > > ### W2: Whether the gains come mainly from stronger models or from the framework itself
> > >
> > > Thank you for this suggestion. To more directly address the concern that the gains may mainly come from stronger annotators rather than our framework, under the limited rebuttal timeline, we conducted an additional controlled experiment on 10k training instances, sampled from the same input pool while preserving the same 1:2:1 ratio of 3 modes as in our main training recipe.
> > >
> > > We define the three settings as follows:
> > > - **Setting A**: Original 10k subset from our training data with human verification
> > > - **Setting B**: Same inputs with Gemini-generated trajectories, followed by Qwen3VL-235A22B (Qwen3VL-A22B) verification
> > > - **Setting C**: Same inputs, but replaces Gemini with Emu3.5-generated trajectories, also followed by Qwen3VL-A22B verification
> > >
> > > In this way, **A vs. B** mainly isolates the effect of **human vs. model-based verification**, while **B vs. C** mainly isolates the effect of the **commercial annotator vs. an open-source annotator**.
> > >
> > > | Setting (Annotation / Verification) | GenEval | KRIS | Omni |
> > > |---|---:|---:|---:|
> > > | Emu3.5 | 0.86 | 73.75 | 8.82 |
> > > | A: Original 10k subset / Human | 0.87 | 76.74 | 9.02 |
> > > | B: Gemini / Qwen3VL-A22B | 0.88 | 76.03 | 8.90 |
> > > | C: Emu3.5 / Qwen3VL-A22B | 0.87 | 76.36 | 8.97 |
> > >
> > > Here, Qwen3VL-A22B and Emu3.5 are both **open-source**, **non-commercial** models. Therefore, Setting C removes both the reliance on a commercial annotator and the reliance on human verification. These comparisons support two points:
> > >
> > > **1. A vs. B (Gemini + human verification vs. Gemini + model verification)**
> > >
> > > This comparison shows that human verification mainly improves rigor and conservativeness, but it is **not necessary** for obtaining strong gains. Even without human verification, model-only verification already preserves a meaningful quality threshold and still delivers substantial improvement.
> > >
> > > **2. B vs. C (Gemini + model verification vs. Emu3.5 + model verification)**
> > >
> > > This comparison shows that the gains are **not tied to a specific commercial annotator**. Even when we replace Gemini with an open-source model (Emu3.5), the resulting performance remains competitive. This suggests that the improvement does not primarily come from distilling a single stronger commercial model, but from the structured trajectory design and training framework itself.
> > >
> > > **Broader implications:**
> > >
> > > Once a verification stage is present, whether performed by humans or by a verifier model, the basic quality of the constructed trajectories is already controlled to a reasonable extent. While Gemini has very strong generation capability, our pipeline design makes the Emu3.5-based setting comparable in practice. We used human verification in the original pipeline mainly for strictness, not because the framework fundamentally requires it to work.
> > >
> > > ---
> > >
> > > ### W3: Scalability
> > >
> > > Based on the additional results above in W2, we would like to highlight two conclusions:
> > >
> > > **First, our framework can achieve similar gains without relying on commercial models or human verification.**
> > >
> > > This directly supports that the framework itself has scalability potential, rather than depending intrinsically on a commercial model or manual review.
> > >
> > > **Second, commercial models should not be conflated with poor scalability.**
> > >
> > > When discussing the effect of commercial models on scalability, the relevant factor is **cost**, not whether the model is "commercial" per se. As we discussed in the previous rebuttal, commercial APIs can in practice be cheaper than local generation. Therefore, **commercial models are not the main bottleneck for scalability**. If a commercial model provides lower cost per usable sample, then using it is in fact easier to scale.
> > >
> > > In contrast, the practical bottlenecks are:
> > > - The **generation cost per valid sample**, and
> > > - The **throughput of human verification**
> > >
> > > Among these two, human verification is indeed the slower component. However, the new experiment above shows that replacing human verification with model verification still leads to similar performance. This means that human verification improves rigor, but is not a hard requirement for the framework to scale effectively.
> > >
> > > ---
> > >
> > > We genuinely appreciate the time and effort you have dedicated to engaging with our work. We have made every effort to thoroughly address your remaining concerns in this second round. All clarifications, additional experiments, and qualitative analyses developed during the rebuttal will be incorporated into the final version. If you feel that our responses and updates have satisfactorily resolved your questions, we would be deeply grateful if you might consider reconsidering your recommendation.

---

### Official Review · Reviewer_3rnH · 2026-03-13

**Soundness:** 3
**Presentation:** 3
**Significance:** 3
**Originality:** 3
**Overall Recommendation:** 4
**Confidence:** 3

**Summary:**

The paper proposes a unified framework for interleaved planning and reflection in a multimodal generative model, with the goal of improving the performance on any-to-image generation tasks. To achieve that, the authors construct an interleaved image generation dataset (around 50K examples). It is created by first using an analyzer (Qwen-235B) and a generator (Gemini-3-Pro-Image) to get sequences of interleaved text and images, which simulate direct generation, reflection mode, and multi-step mode. The data is then verified by two human annotators to guarantee the quality.  Based on the dataset, a base multimodal model (here, they use Emu3.5) is trained in a two-stage manner, where the first stage is for supervised finetuning, and the second stage is for reinforcement learning using GRPO. The reward is composed of an outcome reward, a format reward, and a step-wise reasoning reward provided by an Analyzer. An intra-group complexity penalty is also used to avoid over-reasoning. The method is evaluated on a text-to-image generation benchmark (GenEval), an image-editing benchmark (KRIS-Bench), and an anything-to-Image benchmark (OmniContext). The model achieves higher performance than the base model and controlled variations.

**Compliance With Llm Reviewing Policy:**

Affirmed.

**Final Justification:**

My concerns have been addressed. I keep my original score.

**Key Questions For Authors:**

Please refer to the weakness part.

**Limitations:**

The authors do not discuss limitations in the paper. I suggest that the authors discuss more about the limitations and future work. Please check the points in the weakness part as a reference.

**Strengths And Weaknesses:**

## Strengths:

- The paper studies an important question, facilitating reasoning and test-time scaling for multimodal image generation models. The authors made a solid contribution in building an interleaved reasoning dataset and in developing a model trained to do interleaved reasoning and verification when generating the images. To the best of my knowledge, the interleaved part is novel in the multimodal image generation domain.
- The paper is well-written, the method is easy to understand, and the figures communicate the idea well.
- The experiments cover various types of image generation tasks, and the model shows state-of-the-art performance on the benchmarks. The paper also includes ablations on several components, which help clarify the contribution of different parts of the framework.

## Weakness:

- It’s unclear how the outcome reward is actually calculated, like what the exact metrics are used to evaluate instruction following, consistency, quality, and knowledge. The paper also lacks a more detailed ablation or discussion for the role of different rewards, and how to choose the hyperparameters in the reward calculation (like the weights in Eq. 2).
- The evaluation of the text-to-image generation benchmark is limited to GenEval, where the prompts are relatively short. The paper should evaluate at least one other benchmark, like COCO, DPG-Bench, to demonstrate its performance on longer text prompts.
- The paper does not analyze what generation strategies the model actually learns under the proposed framework. Like, it is unclear whether the model adaptively chooses between direct generation and multi-step generation, or whether the behavior collapses to a single dominant pattern. Also, it would be interesting to explore the influence of the three operation modes from the training data and see how it affects the model’s reasoning and generation.

---

> ### Author Rebuttal · Authors · 2026-03-31
>
> ### W1:Reward Calculation
>
> - The four components of the outcome reward follow established evaluation axes from widely adopted image generation benchmarks\[1-3\]. These dimensions are scored by an LMM evaluator following \[1-3\], employing standard judge-style prompts as detailed in **Appendix E**. Therefore, these components are treated as a unified metric rather than being isolated for individual ablation.
> - The coefficients $w_1,\dots,w_4$ in Eq.2 function as fixed normalization factors instead of tunable hyperparameters. Each sub-score is measured on a 1-5 scale and rescaled by $0.2$. These are then averaged across the four dimensions, resulting in $w_1=w_2=w_3=w_4=0.05$. Thus, Eq.2 serves only to define the normalized outcome reward without introducing further variables for reward tuning. We appreciate the comment and will clarify these definitions in the revised manuscript.
> - Regarding the various reward functions, our work utilizes three reward types and a penalty: outcome, format, and step-wise reasoning rewards, alongside a complexity penalty. The original submission includes ablations for these rewards in Table 3, specifically isolating the effects of the step-wise reasoning reward and the complexity penalty. Specifically, the step-wise reasoning reward primarily drives generation quality, while the complexity penalty reduces reasoning steps while maintaining comparable performance.
>
> \[1\] KRIS-Bench: Benchmarking Next-Level Intelligent Image Editing Models.
>
> \[2\] WorldEdit: Towards Open-World Image Editing with a Knowledge-Informed Benchmark.
>
> \[3\] WiseEdit: Benchmarking Cognition- and Creativity-Informed Image Editing
> ### W2:More experiments on complex text-to-image benchmark
>
> - We appreciate the reviewer's suggestion and have addressed this by incorporating additional evaluations. Accordingly, we have included supplementary results for COCO and DPG-Bench in **<https://anonymous.4open.science/r/re16701-5BA0/COCO_DPG.md>**. These results confirm that our method maintains strong performance on longer text prompts. This validates that the training strategy is effective beyond short-prompt scenarios and generalizes well to diverse benchmarks outside of GenEval.
> ### W3:Mode Selection
>
> - Our model is trained on three operation modes with a 1:2:2 ratio for Direct, Reflection, and Multi-step modes within the 50K training set (lines 695-696), ensuring the model avoids bias toward any single generation pattern.
> - In KRIS-Bench, for example, across 4450 test instances, the model utilizes direct generation for 2180 cases, reflection for 1472, and multi-step generation for 798. This distribution illustrates that the learned policy avoids mode collapse, adaptively selecting strategies according to input difficulty and error characteristics. This behavior aligns with our design goal to route straightforward cases to direct generation, refinement-heavy tasks to reflection, and high-complexity prompts to multi-step planning.
> - Regarding the influence of the three operation modes in training data, the impact of these modes is further analyzed in Table 3: removing reflection primarily impairs KRIS performance, while omitting multi-step planning impacts OmniContext more notably, with the integrated approach performing best. These findings indicate that the three modes serve complementary roles rather than any single mode dominating the others.

---

> > ### Author Rebuttal · Reviewer_3rnH · 2026-04-02
> >
> > Thank you for the detailed rebuttal and additional experiments. My concerns regarding W1 and W2 are fully resolved. For W3, the mode distribution statistics on the test set are helpful, but I would encourage the authors to provide more in-depth analysis, for example, showing whether the model's mode selection correlates with input difficulty or complexity, rather than just reporting overall counts. More qualitative case studies would help demonstrate that the model has truly learned an adaptive strategy rather than outputting modes in a fixed pattern. I encourage the authors to incorporate all these updates in the revised manuscript.

---

> > > ### Author Response · Authors · 2026-04-06
> > >
> > > Thank you for your positive feedback and for the helpful suggestion. We are glad that our clarifications and additional experiments have addressed your concerns regarding W1 and W2.
> > >
> > > ---
> > >
> > > Regarding your suggestion on W3, we agree that more qualitative evidence would help better demonstrate that the model learns an adaptive strategy rather than following a fixed pattern. To further support this point, **we identified a visual example with inputs of different complexity, where the model naturally switches among the three reasoning modes: direct generation, self-reflection, and multi-step planning according to the difficulty of the input**, as illustrated in **<https://anonymous.4open.science/r/re16701-2-22B3/selection_correlation.md>**. For a simple local edit (“add a monitor to the table on the left”), the model uses the Direct mode and completes the instruction in a single pass. For a moderately difficult instruction (“reduce the size of the tables by half”), where the initial edit only partially satisfies the request, the model switches to Self-Reflection to identify the missing change and correct it. For a more compositional instruction (“put the two tables together and drape a blue tablecloth over the joined table”), the model adopts Multi-Step Planning, first merging the two tables and then applying the tablecloth. These cases provide qualitative evidence that the model does not follow a fixed mode pattern, but instead selects different reasoning modes according to the structural complexity of the input. We believe this example provides additional evidence that the mode selection behavior is adaptive rather than static.
> > >
> > > ---
> > >
> > > We genuinely appreciate the time and effort you have dedicated to engaging with our work. We have made every effort to thoroughly address your remaining concerns in this second round. All clarifications, additional experiments, and qualitative analyses developed during the rebuttal will be incorporated into the final version. If you feel that our responses and updates have satisfactorily resolved your questions, we would be deeply grateful if you might consider reconsidering your recommendation.

---

### Official Review · Reviewer_E1hH · 2026-03-18

**Soundness:** 2
**Presentation:** 3
**Significance:** 3
**Originality:** 2
**Overall Recommendation:** 4
**Confidence:** 4

**Summary:**

This paper studies the gap between multimodal understanding and visual generation in Unified Multimodal Models. The authors identify two key bottlenecks—attention entanglement when handling complex instructions and limited ability to refine imperfect generations. To address this, they propose a Self-Adaptive Interleaved Visual Reasoner that dynamically switches among three modes: direct generation, reflection-based refinement, and multi-step planning with interleaved image generation. A hierarchical data construction pipeline is introduced to automatically synthesize reasoning trajectories across these modes, producing a 50k training dataset. The model is trained via supervised fine-tuning followed by RL optimization with GRPO and several reasoning-aware rewards. Experiments on multiple benchmarks demonstrate improvements over existing unified multimodal generation models.

**Compliance With Llm Reviewing Policy:**

Affirmed.

**Final Justification:**

N/A

**Key Questions For Authors:**

1.Is the proposed method applicable to other base models besides Emu3.5? If so, does it require additional adaptation (e.g., hyperparameter changes or architectural adjustments) when applied to different backbones?

2.How does the method perform on a broader set of benchmarks beyond those reported in the paper? For example, results on datasets such as WISE or T2I-CompBench would help provide a more comprehensive evaluation.

3.Could the authors provide hyperparameter sensitivity analysis for the coefficients in Eq. (2), as well as RL training curves? In particular, it would be interesting to see whether the training exhibits behaviors similar to the “aha-moment” phenomenon reported in recent RL-based reasoning models. Additionally, scale-up ablations (e.g., larger datasets or longer training) would further clarify the robustness of the approach.

**Limitations:**

yes

**Strengths And Weaknesses:**

Strengths

1.Unified multimodal models are a highly active research direction in the community, where balancing multimodal understanding and visual generation is a crucial challenge. This paper targets this important problem by studying the gap between instruction understanding and image synthesis.The proposed benchmark is reasonably constructed, with tasks covering diverse domains to reflect real-world variety.

2.The paper is generally well written, with clear figures and tables, making it easy to follow and understand.

3.With carefully designed data construction and training strategies, the proposed method achieves noticeable improvements on three benchmarks: GenEval, KRIS-Bench, and OmniContext.

Weaknesses:

1.The experiments are primarily conducted using Emu3.5 as the backbone. It would be valuable to evaluate the proposed approach on additional base models(Bagel/BLIP3-o/Show-o2) to assess whether the method generalizes across different architectures or requires additional adaptation. Such analysis would further strengthen the scalability and general applicability of the proposed framework.

2.Evaluation is limited. It would be helpful to include results on additional benchmarks, such as WISE and T2I-CompBench, to provide a more comprehensive assessment. In particular, it would be interesting to examine whether the proposed training strategy affects the model’s general multimodal capabilities, which could be evaluated using standard VLM benchmarks.

3.Equation (2) introduces four hyperparameters, but the paper does not provide sensitivity analysis for them. This makes it difficult to determine whether the reported improvements mainly come from the proposed methodology or from careful hyperparameter tuning. This also raises a related concern about whether the method would require model-specific tuning when applied to other base models (W1).

4.Balancing generation and understanding capabilities has been a long-standing and important problem in the community. I would like to see clearer comparisons with related works that make similar claims about addressing the understanding–generation gap and also emphasize reflection-style reasoning. Relevant examples include IRG[1], DraCo[2], Unicorn[3], and TwiG[4]. At minimum, qualitative discussion and proper citations of these approaches would help better position the contribution of this work.

[1] Interleaving reasoning for better text-to-image generation. ICLR 2026.

[2] DraCo: Draft as CoT for Text-to-Image Preview and Rare Concept Generation. arXiv:2512.05112.

[3] UniCorn: Towards Self-Improving Unified Multimodal Models through Self-Generated Supervision. arXiv:2601.03193.

[4] Thinking-while-generating: Interleaving textual reasoning throughout visual generation. arXiv:2511.16671.

---

> ### Author Rebuttal · Authors · 2026-03-31
>
> ### W1-Q1:Extra experiments on other base model
> - We extended our training data and methodology to BAGEL without modifying hyperparameters or architectural configurations. Results in **<https://anonymous.4open.science/r/re16701-5BA0/BAGEL_based_reasoning_variant.md>** show consistent improvements over baseline BAGEL and its reasoning-based variants, indicating that the proposed approach is largely architecture-agnostic rather than dependent on a specific backbone.
>
> ### W2-Q2:Extra experiments on other benchmarks
> - We appreciate the suggestion and have accordingly incorporated experiments on WISE and T2I-CompBench. The results of these additional evaluations are provided in **<https://anonymous.4open.science/r/re16701-5BA0/WISE_CompBench.md>**. These findings demonstrate that our method exceeds baseline performance on both benchmarks, further validating the versatility of our approach.
> - In the original submission, we centered our evaluation on X2I benchmarks because the base model, Emu3.5, is designed as a unified world model rather than a general-purpose VLM. Therefore, our method specifically optimizes generation-time reasoning and editing fidelity rather than standard multimodal tasks. To examine if our training compromises general multimodal comprehension, we evaluate the BAGEL-based variant in **W1** on two standard VLM benchmarks (**<https://anonymous.4open.science/r/re16701-5BA0/BAGEL_VLM.md>**). We do not find obvious degradation after applying our training recipe. We attribute this to the fact that our approach does not optimize generation in isolation; rather, tasks such as planning and reflection inherently necessitate and preserve multimodal understanding.
>
> ### W3-Q3:Coefficients in Eq.2 and RL dynamics
> - The coefficients $w_1,\dots,w_4$ in Eq.2 function as fixed normalization factors instead of tunable hyperparameters. Each sub-score is measured on a 1-5 scale and rescaled by $0.2$. These are then averaged across the four dimensions, resulting in $w_1=w_2=w_3=w_4=0.05$. Thus, Eq.2 serves only to define the normalized outcome reward without introducing further variables for reward tuning. We appreciate the comment and will clarify these definitions in the revised manuscript.
> - Regarding RL dynamics, we present training curves in **<https://anonymous.4open.science/r/re16701-5BA0/RL_training_curves.md>**. We do not observe a discrete phase transition typical of an “aha-moment.” This is likely because our policy is initialized via SFT, which establishes a stable format and basic reasoning capabilities; thus, RL primarily refines strategy selection and efficiency instead of relying on unconstrained exploration. Consistent with this, the curves show steady performance gains alongside a reduction in generation steps, suggesting the model learns efficiency rather than simply increasing reasoning length.
> - Regarding scale-up ablations (larger datasets or extended training), we recognize their value. While constraints during the rebuttal period prevent their inclusion here, we will investigate these factors in subsequent experiments and specify this limitation in the final version.
>
> ### W4:More comparison and discussion with recent works
> - We appreciate the references and will incorporate these studies into the revised manuscript to better contextualize our contributions. Specifically:
>   - IRG [1] explores T2I through a static think-generate-reflect pipeline and a specialized training procedure for interleaved text-image trajectories.
>   - DraCo [2] utilizes a visual CoT approach by replacing abstract textual planning with low-resolution drafts, followed by semantic verification and selective correction.
>   - UniCorn [3] presents a self-improvement methodology for multimodal models, primarily targeting T2I, where a single model assumes the roles of proposer, solver, and judge to generate self-supervision.
>   - TWIG [4] integrates textual reasoning within the patch-level visual generation process instead of restricting it to pre- or post-generation phases.
> - In contrast, our work addresses the broader X2I setting. By identifying two primary failure modes, attention entanglement and visual refinement, we move beyond a fixed reasoning sequence to an adaptive policy that selects between direct generation, structured reflection, or multi-step planning based on task complexity.
> - Furthermore, existing methodologies typically rely on rigid data templates: e.g., decomposed T2I interleaving in IRG [1], draft-correction pairs in DraCo [2], self-play self-supervision in UniCorn [3], or fixed region-wise interleaving in TWIG [4]. Our dataset is developed through a hierarchica process that routes instances into appropriate strategies, verified by human annotators to ensure high-quality reasoning trajectories.
> - As demonstrated in Table 5 and **<https://anonymous.4open.science/r/re16701-5BA0/WISE_CompBench.md>**, our method consistently outperforms these baselines across the GenEval, WISE, and T2I-CompBench benchmarks.

---

> > ### Author Rebuttal · Reviewer_E1hH · 2026-04-04
> >
> > Thank the authors for their response. My concerns have been addressed.

---

> > > ### Author Response · Authors · 2026-04-06
> > >
> > > We genuinely appreciate the time and effort you have dedicated to engaging with our work. All clarifications, additional experiments, and qualitative analyses developed during the rebuttal will be incorporated into the final version.

---

### Decision · Program_Chairs · 2026-04-30

**Decision:**

Accept (regular)

**Comment:**

The original reviews of this submission include one reject and three weak rejects. The reviewers’ main concerns focus on experiments and limited evaluation, hyperparameter analysis, reward design, data scalability, presentation, etc. The authors provided very detailed responses to the reviewers’ original comments and the follow-up questions in the rebuttal process.  Most of the reviewers’ concerns are addressed. As a result, Reviewer E1hH and Reviewer fTr6 raised their scores from 3 to 4, and Reviewer 3rnH and Reviewer RbLL kept their positive scores of 4. The ACs agree to accept this paper.